# The Dual-Route Model of Induction

**Sheridan Feucht, Eric Todd, Byron Wallace, & David Bau** *
Northeastern University
{feucht.s,todd.er,b.wallace,d.bau}@northeastern.edu

## Abstract

Prior work on in-context copying has shown the existence of *induction heads*, which attend to and promote individual tokens during copying. In this work we discover a new type of induction head: *concept-level* induction heads, which copy entire lexical units instead of individual tokens. Concept induction heads learn to attend to the ends of multi-token words throughout training, working in parallel with token-level induction heads to copy meaningful text. We show that these heads are responsible for semantic tasks like word-level translation, whereas token induction heads are vital for tasks that can only be done verbatim (like copying nonsense tokens). These two "routes" operate independently: we show that ablation of token induction heads causes models to paraphrase where they would otherwise copy verbatim. By patching concept induction head outputs, we find that they contain language-independent word representations that mediate natural language translation, suggesting that LLMs represent abstract word meanings independent of language or form.

## 1 Introduction

How do language models repeat sequences of tokens in-context? Imagine that you are asked to copy a random string of characters from a leaflet of paper into a notebook: `'oane dnn t ephzawfeew eausr lthii'`. This would be an achievable but laborious task, requiring careful attention to each subsequent character. Now, imagine instead that these characters are rearranged into the phrase `'the false azure in the windowpane'`. There are now two ways of copying the sequence: character-by-character, or by leveraging your understanding of English to copy large swaths of characters at a time.

LLMs can copy text using *induction heads*, a type of circuit found in decoder-only transformer models that enables them to copy sequences in-context (Elhage et al., 2021; Olsson et al., 2022). Given the sequence `w|ax|wing...w`, an induction head at the second occurrence of `w` would attend to and promote the earlier token `ax` by scanning for previous token information. Prior work has argued that induction heads are responsible for broader in-context learning capabilities (Olsson et al., 2022). However, it is unclear how attention heads operating on a token-by-token basis might handle "fuzzy" copying tasks like translation, which often requires conversion between words of differing token lengths (e.g., `p|ommes| de| terre` → `pot|atoes`). In this paper, we show that LLMs use *concept induction heads* in parallel with token induction heads to copy meaningful text.

**The Dual-Route Model of Induction.** Psychologists who study reading in the brain describe two parallel routes through which people read: a sublexical route that converts letter strings into speech sounds, and a lexical route through which word meanings can be directly accessed as entire units (Marshall & Newcombe, 1966; 1973; Dehaene, 2009). If the sublexical route is damaged, patients exhibit a condition known as *deep dyslexia*, where they can understand the meanings of words without being able to access their sound or spelling: Marshall & Newcombe (1966) describe a patient who, after a brain injury, would read the word CANARY as "parrot" and COLLEGE as "school," indicating that he could still access word meanings, but was unable to read individual graphemes.

---

*Code and data available at https://dualroute.baulab.info.

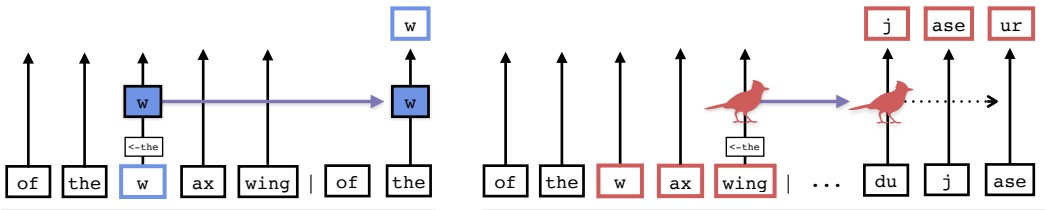

**Token Induction:** *verbatim copying*          **Concept Induction:** *semantic copying*

Figure 1: The dual-route model of induction. LLMs develop token induction heads, which are used for verbatim copying, alongside concept induction heads, important for translation and "fuzzy" copying tasks. These two routes work in parallel to copy meaningful text.

In this work, we consider the possibility of an analogous dual-route model of induction in LLMs, where tokens are equivalent to graphemes. Given a piece of meaningful text, models can either copy by shifting individual subword tokens, or by accessing detokenized lexical information over an entire span of tokens. We characterize two types of induction:

**Token Induction.** Figure 1 illustrates token-level induction circuits discovered in previous work. Elhage et al. (2021) show that in two-layer transformers, models load previous token information into each hidden state, which allows induction heads in future layers to attend to these states and copy the corresponding token. We design a causal intervention to identify token induction heads in Section 2. When these heads are ablated, models lose the ability to copy sequences verbatim (Section 4), exhibiting "symptoms" analogous to patients with deep dyslexia (Hinton & Shallice, 1991; Zorzi et al., 1998).

**Concept Induction.** Figure 1 also depicts our current understanding of *concept induction* heads, which are responsible for copying lexical information. To isolate these heads, we define a *concept copying* score based on causal mediation for a simple multi-token copying task (Section 2). Instead of attending to the next token, we find that these heads attend to the ends of multi-token words (Section 3), where concept information is more likely to be stored (Meng et al., 2022; Geva et al., 2023; Nanda et al., 2023; Feucht et al., 2024; Kaplan et al., 2025). Concept induction heads are vital for semantic copying tasks (Section 4), and output word representations that are language-agnostic (Section 6).

## 2  Token & Concept Copying Scores

### 2.1  Approach

To identify concept induction heads, we search for attention heads responsible for copying multi-token words by measuring the effects of causal interventions (Vig et al., 2020; Geiger et al., 2023). We hypothesize that if a head increases the probability of future tokens for multi-token concepts, it is actually copying the *entire* concept.

Figure 2 illustrates our approach. We first sample random tokens[1] to create an induction prompt $x_1 x_2 ... x_n | x_1' x_2' ... x_n'$, using the newline token as a separator between the first and second repeating occurrences of $x_1 ... x_n$. Then, we append a single concept made of $m$ tokens $c_1 ... c_m$ to each half of the repeated sequence $s$. These concepts are sampled from a set of multi-token concepts $\mathcal{C}$, with lengths uniformly distributed over $2 \leq m \leq 5$. Prompts are truncated so that the final token is always $c_1'$ regardless of $m$. We set $n = 30 - m$.

This yields a clean prompt $p_{clean} = x_1 x_2 ... x_n c_1 ... c_m | x_1' x_2' ... x_n' c_1'$, which we corrupt by replacing the first half with different random tokens: $p_{corrupt} = y_1 y_2 ... y_{n+m} | x_1' x_2' ... x_n' c_1'$. We patch the activations of each attention head $a^{(l,h)}$ (layer $l$, head $h$) from the penultimate token

---

[1]To avoid confounds from undertrained tokens (Land & Bartolo, 2024), we sample tokens by randomly selecting a document from the Pile (Gao et al., 2020), tokenizing it, and shuffling the token order. We adopt this approach for all random token sampling henceforth.

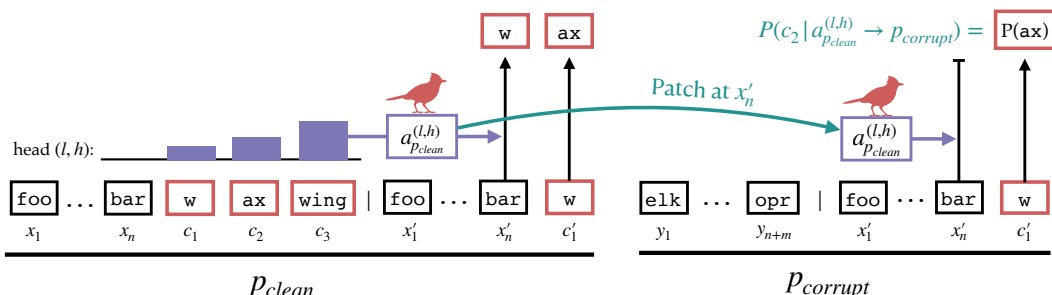

Figure 2: We patch the outputs of each attention head from `bar` in $p_{clean}$ to `bar` in $p_{corrupt}$ to see whether that head has an impact on the "next-next" token $P(c_2)$, which is $P(\text{ax})$ in this example. Our hypothesis is that the heads that increase $P(\text{ax})$ in this setting actually carry the entire concept of "waxwing." See Section 2.1 for notation details.

| Model | Hugging Face ID | $|\mathcal{V}|$ | $t$ | Citation |
|---|---|---|---|---|
| Llama-2-7b | `meta-llama/Llama-2-7b-hf` | 32k | 2T | Touvron et al. (2023) |
| Llama-3-8b | `meta-llama/Meta-Llama-3-8B` | 128k | 15T+ | Grattafiori et al. (2024) |
| OLMo-2-7b | `allenai/OLMo-2-1124-7B` | 100k | 4T | OLMo et al. (2025) |
| (OLMo-2-1b) | `allenai/OLMo-2-0425-1B` | 100k | 4T | OLMo et al. (2025) |
| Pythia-6.9b | `EleutherAI/pythia-6.9b` | 50k | 0.3T | Biderman et al. (2023) |

Table 1: Models used in this paper. $|\mathcal{V}|$ is the model's token vocabulary size, and $t$ is the number of tokens the model was trained on (in trillions). We evaluate OLMo-2-1b on only a subset of experiments.

position of $p_{clean}$ (i.e., from $x'_n$) into the same position in $p_{corrupt}$. Then, the concept copying score for head $(l, h)$ over concept set $\mathcal{C}$ is defined as

$$\text{CONCEPTCOPYING}(l, h) = \frac{1}{|\mathcal{C}|} \sum_{c \in \mathcal{C}} \left( P(c_2 | a^{(l,h)}_{p_{clean}} \xrightarrow{x'_n} p_{corrupt}) - P(c_2 | p_{corrupt}) \right) \qquad (1)$$

where $\xrightarrow{x'_n}$ indicates that activations $a^{(l,h)}$ are being patched into the $p_{corrupt}$ context at the position corresponding to $x'_n$. Because $c_2$ is predicted at the token position *after* $x'_n$, we are measuring increase in probability for the "next-next token" (Pal et al., 2023; Wu et al., 2024). Our hypothesis is that if an attention head increases probability for the future token $c_2$, this may be because it is carrying information about the entire concept $c_1...c_m$.

We also want to find attention heads that are responsible for copying one token at a time (i.e., token induction heads). We run the same procedure to find these heads, with two differences: (1) We use random tokens $r_1...r_m \in \mathcal{R}$ instead of concepts, and (2) we measure the impact of each head on $P(r_1)$, instead of $P(r_2)$. This gives us our token copying score:

$$\text{TOKENCOPYING}(l, h) = \frac{1}{|\mathcal{R}|} \sum_{r \in \mathcal{R}} \left( P(r_1 | a^{(l,h)}_{p_{clean}} \xrightarrow{x'_n} p_{corrupt}) - P(r_1 | p_{corrupt}) \right) \qquad (2)$$

For this experiment and for Section 3, we take concepts from the COUNTERFACT dataset (Meng et al., 2022), which consists of subject-object relations (e.g., "Paul Chambers plays bass"). We sample a subset of subjects from this dataset to use as our set of concepts $\mathcal{C}$, where $|\mathcal{C}| = 1024$. We could also use generic multi-token words, but results from Feucht et al. (2024) suggest that models treat both types of sequences similarly. We also measure scores for $P(c_1)$ and $P(r_2)$, but do not focus on them in this work (see Appendix A).

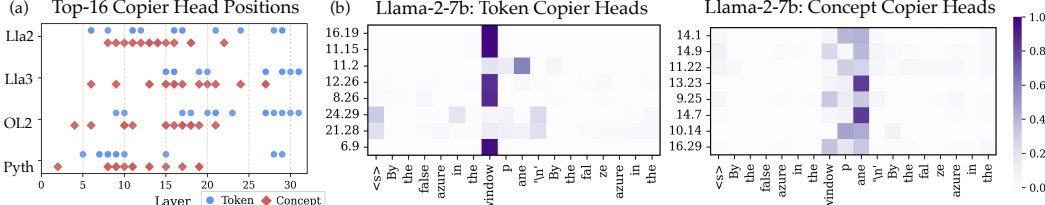

Figure 3: We use causal mediation to identify *token copier heads* that copy the next random token, as well as *concept copier heads* that copy future concept tokens. (a) Distribution of the top-16 token and concept copier heads across model layers. (b) Value-weighted attention scores over a repeating phrase for the top causally-ranked heads ($l.h$) in Llama-2-7b. At the final token position "the", token copier heads attend to the next token ("**window**.p.ane"), whereas concept copier heads attend to the end of the next word ("window.p.**ane**").

## 2.2 Results

We calculate causal scores for all heads in each model[2] in Table 1. This gives us two ways of ranking heads in a model: by concept copying score and by token copying score. Figure 3a shows that concept copier heads seem to be concentrated in mid-early layers, whereas token copier heads are more sporadic, and are more likely to appear at late layers. We find that there is little overlap between the top-*k* heads of each ranking (Figure 32) and that token & concept scores are not correlated (Appendix C.2), implying that these are two separate roles.

As a baseline, we also calculate increases in probability for future random tokens $P(r_2)$ when patching individual heads (Appendix A). This gives a sense of whether heads are copying several arbitrary tokens at a time. While some heads do increase the probability of future random tokens, their maximum intervention effects are at least three times smaller than heads that promote future entity tokens (for all models). We take this as preliminary evidence that concept copying heads copy meaningful units, not just arbitrary lists of tokens.

We also calculate copying scores throughout training checkpoints for OLMo-2-7b and Pythia-6.9b. Some of the top concept induction heads in OLMo-2-7b have high token copying scores before they develop into concept induction heads, suggesting that concept induction heads might develop from token induction heads. However, this pattern is not as clear for Pythia-6.9b. We show examples of head trajectories throughout training in Appendix B.

## 3 Next-Token & Last-Token Attention Scores

Where do concept copying heads attend? In previous work, Olsson et al. (2022) found that token induction heads always attend to the next token—the one that they are about to copy. If concept copier heads transfer entire concepts at once, we would expect them to attend to the ends of multi-token words, where concept information is usually stored (Meng et al., 2022; Geva et al., 2023; Nanda et al., 2023). Figure 3b shows an example of attention patterns for both types of heads in Llama-2-7b. When the model is about to copy a multi-token word, its token copier heads attend to the next token ("**window**.p.ane"), while its concept copier heads attend to the end of the next word ("window.p.**ane**").

### 3.1 Approach

To capture this attention behavior, we design a *last-token matching score* which measures how much attention is paid to the last token of a multi-token concept. First, we select a concept $c = c_1...c_m$ from COUNTERFACT subjects $\mathcal{C}$, evenly sampling across concept lengths $2 \leq m \leq 5$. We then construct random repeated sequences of tokens as before (see §2) and insert the concept at the end of the first half to create the prompt $x_1 x_2...x_n c_1...c_m | x'_1 x'_2...x'_n$. The last-token matching score is then calculated as an average over the concept set $\mathcal{C}$:

---

[2]We use the Hugging Face (Wolf et al., 2020) implementation of each model.

$$\text{LASTTOKENMATCHING}(l, h) = \frac{1}{|\mathcal{C}|} \sum_{c \in \mathcal{C}} \left( A^{(l,h)}[x'_n, c_m] \right) \tag{3}$$

where $A^{(l,h)}$ represents the value-weighted attention scores for head index $h$ at layer $l$, and the square brackets indicate that we collect attention paid from $x'_n$ to $c_m$.

For comparison, we also calculate attention paid to the next token over random sequences, known in previous work as "prefix-matching" scores (Olsson et al., 2022; Bansal et al., 2023). In this work, we refer to this as a head's *next-token matching score.* Just like in §2, we use the same procedure that we do for last-token matching scores, with two differences: (1) we replace concepts with random spans of tokens $r = r_1...r_m$, and (2) we calculate attention paid to the *next* random token $r_1$, instead of the last token.

$$\text{NEXTTOKENMATCHING}(l, h) = \frac{1}{|\mathcal{R}|} \sum_{r \in \mathcal{R}} \left( A^{(l,h)}[x'_n, r_1] \right). \tag{4}$$

Following Kobayashi et al. (2020), all attention scores in this work are calculated using value-weighting; i.e., we multiply attention weights for each token by the $L^2$ norm of their value vectors, then renormalize so the scores sum to one. This tends to account for cases where attention "rests" on unimportant tokens like  at the beginning of a prompt.

## 3.2 Results

Figure 4 shows next-token and last-token matching scores for heads of each type in Llama-2-7b, averaged over 2048 COUNTERFACT entities. Token copier heads tend to have high next-token matching scores, whereas concept copier heads tend to have high last-token matching scores.[3] We show only the top 16 heads (ranked using the scores defined in Equations 1 and 2) for readability, but include top-64 scores for all models in Appendix C.1.

We also calculate correlations between causal and attention-based scores. As expected, token copying scores positively correlate with next-token matching scores (r=0.63, p<0.001) whereas concept copying scores correlate with last-token matching scores (r=0.44, p<0.001) for Llama-2-7b. We find significant correlations for all models; see Appendix C.2.

Some heads seem to attend to both next-tokens and last-tokens. In Llama-2-7b, head 11.2 is the third-highest token copier head, but also has a relatively high last-token copying score (and attends to ane in Figure 3b). We also see concept copier heads with next-token matching scores up to 0.20, such as head 13.23. This suggests that some heads may just copy the next "thing," whether that be the next token or the next concept.

## 4 Lesioning Concept and Token Copier Heads

We have found a set of attention heads that attend to the ends of multi-token entities (Section 3) and help to copy the second tokens of those entities (Section 2). We hypothesize that these are *concept induction heads*, which copy by transferring entire concept representations at once. In this section, we show through targeted ablations that concept heads are vital for "fuzzy copying" tasks that deal with lexical semantics, whereas token induction heads are important for verbatim copying tasks.

## 4.1 Approach

First, we define a new "vocabulary list" task that requires models to copy a list of words in-context. Using word-pair data from Conneau et al. (2017) for five languages, we enumerate $n = 10$ words, followed by a parallel list of English translations for each word (see Appendix D.2 for full examples). Using data from Nguyen et al. (2017), we also build similar

---

[3]Last-token attention scores are lower overall because attention can spread over the length of the concept; for example, "window.p" could be treated as a concept if the model guesses that the word is "windowpane" (we can already see this for some heads in Figure 3b).

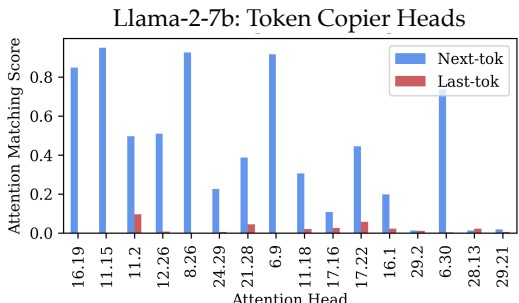 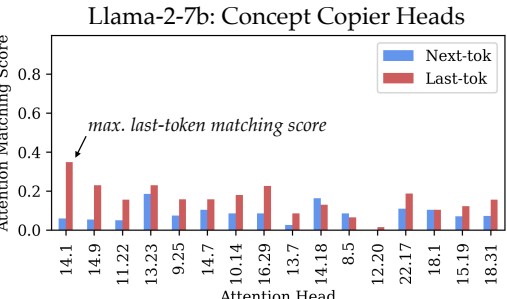

Figure 4: Attention-based matching scores for the highest-scoring causal heads in Llama-2-7b. Consistent with prior work, heads that are responsible for copying the next random token have high next-token matching scores. On the other hand, heads that are responsible for promoting future entity tokens have the highest last-token matching scores.[3]

prompts for uppercased and title-cased English words, synonyms, and antonyms. We calculate first-token accuracy and exclude word pairs that have the same first token. Finally, we add two verbatim tasks, where the first and second list are identical: English copying, using words from Conneau et al. (2017), and "nonsense copying," using randomly-sampled tokens. Models are evaluated on 1024 prompts per task.

For these tasks, we mean-ablate (Wang et al., 2023a) sets of concept and token induction heads to observe impact on model performance. We use causal scores from Section 2 to rank heads in each model twice: once according to concept copying score, and once according to token copying score. We ablate the top-$k$ heads in each of these rankings by replacing their activations across all token positions with their mean activations on random Pile documents ($n = 1024$).

## 4.2 Results

Figure 5 shows ablation results for Llama-2-7b. We see two separate "routes" through which models can copy: a semantic route, via concept induction heads (heads with high concept copying scores), and a verbatim route, using token induction heads. When concept induction heads are ablated, we see a drop in translation, synonym, and antonym accuracy, with no effect on surface-level tasks. When token induction heads are ablated, nonsense copying fails, but the model can still use concept heads to complete the rest of the tasks. English copying remains unaffected for both types of ablation (e.g. pea|coat → pea|coat), as it can be solved using either type of induction. Like English copying, uppercasing tasks can also be done semantically or without regard to meaning. We report similar results for other models in Appendix D. For all models, ablation of concept induction heads damages semantic tasks much more than it does surface-level tasks. Llama-3-8b and OLMo-2-7b behave similarly to Llama-2-7b, but token ablation results for Pythia-6.9b are less clear-cut.

In Appendix D.3, we repeat this experiment while controlling for word length to test whether we are simply distinguishing between multi- and single-token tasks. For single-token words, we see the same pattern, with a weaker (but still distinct) separation between translation and verbatim copying when ablating concept heads. Thus, in cases where words are constrained to a single token, token heads may also be able to assist in semantic tasks.

**Qualitative Examples.** To get a sense of how model outputs look when token induction heads are ablated, we prompt token-ablated models to copy entire sentences verbatim. Ablating a small number $k$ of token attention heads results in "paraphrasing" behavior instead of copying behavior; in other words, the model is able to copy the *meaning* of the sentence, but doesn't get every token exactly right. Box 4.1 shows that Llama-2-7B starts to paraphrase when $k = 32$ token induction heads are ablated.[4] We find that $k = 32$ is a

---

[4]To encourage copying, we feed the sequence twice; we omit the first occurrence here for brevity.

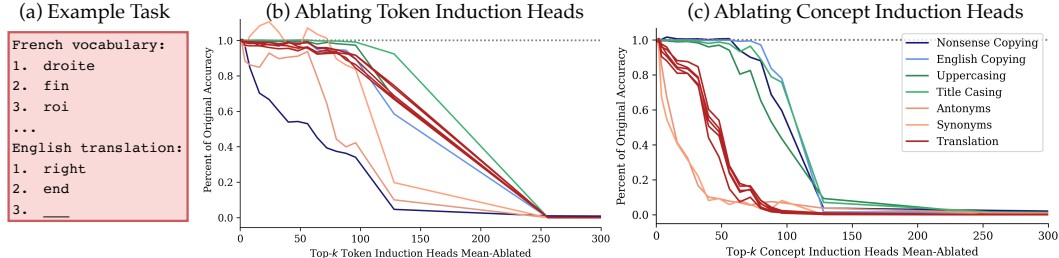

Figure 5: Mean-ablating the top-$k$ copier heads for "vocabulary list" tasks in Llama-2-7b. Ablating token copier heads damages copying for nonsense tokens, without affecting tasks that can be performed semantically. Ablating *concept* copier heads damages performance for translation, synonyms, and antonyms, without affecting surface-level copying. English copying, which can be done either semantically or token-by-token, remains high for both types of ablation, as do uppercasing tasks (which can be done without regard to semantics). We plot results relative to models' original task accuracies; see Appendix D for details.

sweet spot that allows us to observe this effect without causing the model to stop copying altogether. Box 4.2 shows ablated Llama-3-8b generations when prompted to copy a piece of Python code. When token induction heads are ablated, the model still copies the meaning of the original snippet, but writes it using list comprehension. We provide examples of this paraphrasing behavior for all models in Appendix D.4.

---

**Box 4.1: (Llama-2-7b) Original Model vs. Top-32 Token Induction Heads Ablated**

I have reread, not without pleasure, my comments to his lines, and in many cases have caught myself borrowing a kind of opalescent light from my poet's fiery orb.
I **have reread, not without pleasure, my comments to his lines, and in many cases have caught myself borrowing a kind of opalescent light from my poet's fiery orb.**

- - - - - - - - - - - - - - - - - - - - - - - - - - - - - - - - - - - - - - - - - - - -

...
I **have reread my comments on his lines, and I have caught myself many times borrowing from his fiery orb a kind of opalescent light.**

---

**Box 4.2: (Llama-3-8b) Original Model vs. Top-32 Token Induction Heads Ablated**

```
foo = []
for i in range(len(bar)):
    if i % 2 == 0:
        foo.append(bar[i])
foo = []
for i in range(len(bar)):
    if i % 2 == 0:
        asdf.append(bar[i])
```

```
foo = []
for i in range(len(bar)):
    if i % 2 == 0:
        foo.append(bar[i])

foo = [bar[i] for i in range(
len(bar)) if i % 2 == 0]
```

---

## 5 Concept Heads Reveal Semantics of Hidden States

If concept induction heads work with meaningful representations, the subspaces that they read from and write to must contain semantic information. What do these subspaces look like? We use the attention weights of concept heads to define a *concept lens* $L_{C_k} \in \mathbb{R}^{(d,d)}$ that reveals the semantic information contained within any particular hidden state. Specifically, we sum the OV matrices (Elhage et al., 2021) of the top-$k$ concept induction heads $C_k$ to obtain $L_{C_k}$. We then apply this transformation to a hidden state, followed by the model's final normalization and decoding head (see Appendix E for details).

We can use this linear transformation to visualize how a model represents a word in a particular context. For example, consider the word *cardinals*, which could refer to a sports

| Prompt | Concept Lens Outputs ($l = 20$) |
|---|---|
| cardinals | ' Card', ' cardinal', 'Card' |
| he was a lifelong fan of the cardinals | ' Card', ' football', ' baseball', |
| the secret meeting of the cardinals | ' Rome', ' Catholic', ' Card' |
| in the morning air, she heard northern cardinals | ' birds', ' bird', ' Bird', |

Table 2: Applying concept lens to hidden states for the token `inals` in the word *cardinals* reveals context-sensitive semantic representations. We show layer $l = 20$ with weights from the top $k = 80$ concept induction heads for Llama-2-7b. See Appendix E.

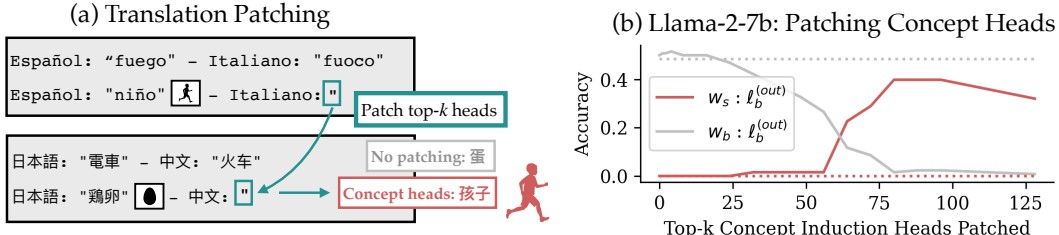

Figure 6: (a) Patching the top-$k$ concept induction head outputs from a Spanish-Italian prompt into a Japanese-Chinese prompt. (b) Patching concept induction heads changes the concept output by the model without affecting the language (i.e., *niño*, the Spanish word for "child," is translated into Chinese instead of Italian). This effect is strongest for $k = 80$, which is also where the largest separation between semantic and literal copying performance is found in Figure 5c. Across $n = 128$ examples, this approach causes the model to output the source Spanish word in the base output language with an accuracy of about 0.40 (solid red line). This is comparable to the model's original Japanese-Chinese translation accuracy of 0.48 (dotted gray line).

team, senior members of the Catholic church, or a group of red birds. By transforming the hidden state for the token `inals` with $L_{C_k}$ and projecting to vocabulary space, we can see that Llama-2-7b has a different semantic representation for *cardinals* depending on the context that the word is in, further supporting the idea that concept heads represent word *meanings* rather than surface-level token information.

## 6 Concept Induction is Language-Agnostic

If concept induction heads are important for translation tasks, what do they output? We hypothesize that the representations being copied by concept induction heads are abstract representations of word *meanings*. In other words, we posit that concept induction heads have the same activations when copying "waxwing" as they do when copying "свиристель," as these two words refer to the same concept, and are only expressed differently on a surface level. This is in line with previous work suggesting that concept information in LLMs may be language-agnostic (Wendler et al., 2024; Dumas et al., 2024; Brinkmann et al., 2025), though Schut et al. (2025) argues concepts for some tasks may be biased towards English. For model representations to be truly language-agnostic, they must be trained on those languages—thus, even if high-resource languages are represented in a unified semantic space, this effect may not hold for low-resource languages. Unfortunately, as we do not evaluate on low-resource languages in this work, we cannot make claims as to how models represent low-resource languages.

### 6.1 Approach

We adopt a similar experimental approach to Dumas et al. (2024). They provide a dataset of word-level translation prompts, where the model is shown five $(\ell^{(in)}, \ell^{(out)})$ pairs and prompted to translate a word $w$ from $\ell^{(in)} \to \ell^{(out)}$. Following their approach, we define a

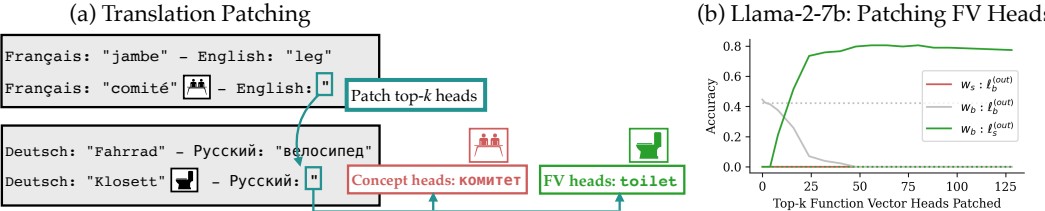

Figure 7: (a) Patching concept heads changes semantics without affecting language, while patching FV heads changes output language without affecting meaning. In red, we show the same experiment from Figure 6: here, patching concept heads causes the model to output the source concept "committee" in Russian. In green, we show that patching FV heads at the same position for the same prompt causes the model to output the base concept "toilet" in English. (b) Across $n = 128$ examples, patching FV heads from a French-English prompt into a German-Russian prompt flips the output language to English with an accuracy of about 0.80 (green line). This is higher than the model's original German-Russian translation accuracy (gray dotted line, approximately 0.41), perhaps because translating into English is easier for Llama-2-7b than translating into Russian.

source prompt $s : \ell_s^{(in)} \rightarrow \ell_s^{(out)}$ and a base prompt $b : \ell_b^{(in)} \rightarrow \ell_b^{(out)}$, where input and output languages differ between source and base prompts. For example, a source prompt might translate from Spanish to Italian, and a base prompt might translate from Japanese to Chinese. These prompts translate two different words $w_s$ and $w_b$. Their target outputs are those words expressed in their respective output languages: $w_s : \ell_s^{(out)}$ and $w_b : \ell_b^{(out)}$.

We patch the set of activations $a_s^k = \{a_s^{(l,h)} | (l,h) \in C^k\}$ of the top-$k$ concept copying heads $C^k$ from the last token position of $s$ into the last token position of $b$. We generate $t = 10$ tokens in this manner with NNsight multi-token generation (Fiotto-Kaufman et al., 2025), intervening for each newly-generated token. Then, we measure performance by evaluating accuracy of model generations. Specifically, given a generated string $w$, we consider $w$ equal to a ground truth label if it is contained within or contains the ground truth string. Dumas et al. (2024) provide multiple possible translations for output words, and $w$ is marked correct if it is equal to any of these synonyms.

## 6.2 Results

Figure 6 shows results for Llama-2-7b when patching outputs from a Spanish-Italian prompt into a Japanese-Chinese prompt. Patching concept induction heads causes the model to output the source word $w_s$ in the base language $\ell_b^{(out)}$. For example, patching the outputs of concept heads from a Spanish-Italian prompt translating "child" into a Japanese-Chinese prompt causes the model to output the word for "child" in Chinese. This intervention is the most effective at about $k = 80$, which also corresponds to the point where translation and nonsense copying are the most separate in Figure 5c. We show similar results for Llama-3-8b and more language pairs in Appendix F. This suggests that concept induction heads transport semantic representations that are expressible across multiple languages.

# 7 Function Vector Heads Complement Concept Induction Heads

Function vector (FV) heads are attention heads whose outputs help to promote in-context tasks. The outputs of FV heads for a few-shot antonym task, for example, will cause the model to output antonyms when added to new contexts (Todd et al., 2024). We argue that FV heads can be thought of as "sisters" to concept induction heads: in the case of translation, concept heads copy semantic information, whereas FV heads copy *language* information.

We calculate correlations between FV scores and concept copying scores and find significant, albeit weak, positive correlations for all models (Figure 48), suggesting there may be some relationship between these two types of heads. This is perhaps because they can both

be thought of as "soft" induction heads, responsible for copying high-level conceptual information in-context.

However, FV and concept heads seem to play two distinct roles in the tasks we examine. We focus on translation, and patch outputs of FV heads between translation prompts using the same approach that we do for concept induction heads in Section 6. Our experimental setup and data is identical, except we patch activations $a_s^k = \{a_s^{(l,h)} | (l,h) \in F^k\}$, where $F^k$ represents the top-$k$ FV heads for a given model. We then measure the output of the base word in the source output language, $w_b : \ell_s^{(out)}$. In other words, we measure whether patching FV heads changes the output language while retaining the original base concept.

Figure 7 shows that for the exact same prompts, patching FV heads causes the model to output the same concept in a different language, whereas patching concept induction heads causes the model to output a different concept in the same language. The outcome of patching FV heads is more sensitive to output language (i.e., the effect is strongest when the output language is English), but nonetheless suggests that FV heads play a distinct role from concept induction heads (see Appendix G).

We also replicate ablations from Section 4 using FV rankings instead of concept and token copying rankings (Appendix G). Ablation of FV heads damages performance for non-verbatim tasks significantly, which makes sense: models cannot perform a task without knowing what the task is.

## 8 Related Work

**Induction Heads and ICL.** The in-context learning capabilities of LLMs as demonstrated by Brown et al. (2020) motivate a body of research on ICL (Dong et al., 2024; Lampinen et al., 2024). We build on the discovery of Elhage et al. (2021) and Olsson et al. (2022), which characterizes token induction heads. Concurrent with our work, Yin & Steinhardt (2025) argue that function vector (FV) heads (Todd et al., 2024), not token induction heads, are primarily responsible for few-shot ICL performance. Similarly, Yang et al. (2025) find that FV heads, which they call *symbolic induction heads*, perform induction over abstract variables. Akyürek et al. (2024) document *n-gram heads*, an n-gram generalization of induction head behavior, and Ren et al. (2024) study *semantic induction heads* that encode syntactic and semantic relations. Our work differs from these previous studies in that we identify concept induction heads via their ability to copy multi-token concepts.

**Concept Representations in LLMs.** We build upon previous work showing that LLMs contain internal representations of "concepts" beyond individual tokens (Kaplan et al., 2025; Hewitt et al., 2025). Some work has studied how individual tokens are converted into abstract concept representations via neurons (Elhage et al., 2022; Gurnee et al., 2023; Nanda et al., 2023) or attention heads (Correia et al., 2019; Ferrando & Voita, 2024). Other work has shown that this concept-level information is stored at the ends of multi-token entities/phrases across various settings for factual recall (Meng et al., 2022; Hernandez et al., 2024; Feucht et al., 2024; Ghandeharioun et al., 2024; Ferrando et al., 2025), and classification tasks (Wang et al., 2023b; Tigges et al., 2024; Marks & Tegmark, 2024). Our work identifies particular heads that transport this conceptual information between token positions.

## 9 Conclusion

In this work we investigate how LLMs copy lexical information. We find *concept induction heads* that are responsible for copying multi-token words, and whose representations are expressible across languages. These heads act as a second "route" through which models can copy meaningful text, alongside previously discovered token induction heads (Olsson et al., 2022). Concept induction heads are one example of how LLMs might use induction in a general way to transport abstract contextual information between hidden representations.

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

# A    Token & Concept Copying Scores

We provide copying scores for Llama-2-7b in Figure 9 and Figure 10, for Llama-3-8b in Figure 11 and Figure 12, for OLMo-2-7b in Figure 13 and Figure 14, and for Pythia-6.9b in Figure 17 and Figure 18.

Although we calculate intervention scores for the next token over entities ($c_1$) and the next-next token for random tokens ($r_2$), we do not focus on these scores in our work. In the former case, heads that are responsible for promoting the next concept token (i.e., $c_1$) could be either concept induction heads or token induction heads; we assume that this score would not help us to differentiate between the two. The latter score gives us information on heads that copy multiple arbitrary tokens (e.g., $r_2$) at a time (heads that copy "lists" of random tokens). Careful comparison of the right-hand plots for entity tokens versus random tokens (e.g., Figure 9 vs. Figure 10) shows that these heads have weaker effects, which helps to support our hypothesis that concept heads are not just "copying lists of tokens." However, we do not utilize these scores further in this work. In Figure 8, we plot the concept copying score against token copying score for each model. We find that concept and token copying scores are either negatively correlated (Llama-2-7b, OLMo-2-7b, and Pythia-6.9b) or uncorrelated (Llama-3-8b). We take this to mean that concept copying and token copying are two distinct roles.

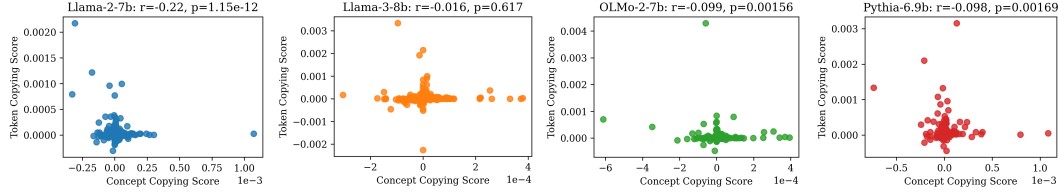

Figure 8: Concept copying scores plotted against token copying scores for every model. These scores are either not correlated or negatively correlated. In other words, there are few heads, if any, that are causally important for both random next-token copying and copying of future concept tokens.

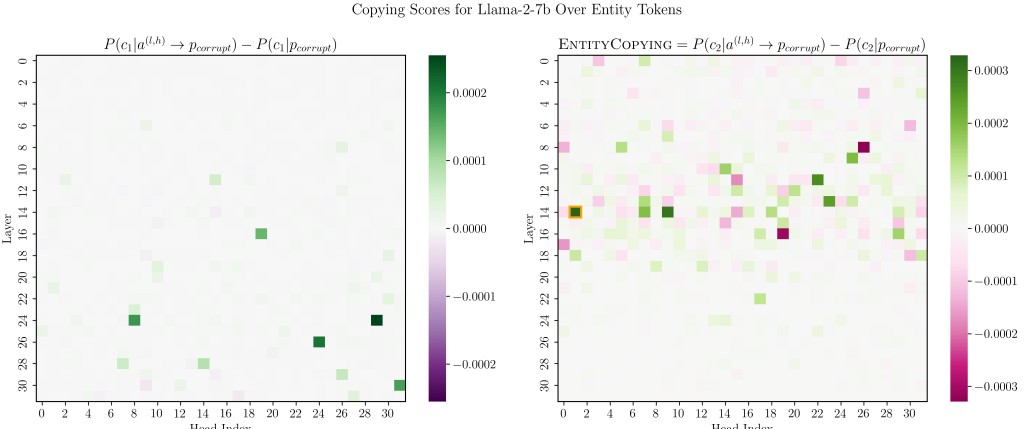

Figure 9: Probability differences when patching head activations for **Llama-2-7b** over **entity tokens**.[*] The right-hand side shows concept copying scores; we do not utilize the left-hand scores in this work. We patch at token position $x'_n$ and measure increase in probability for $c_1$ at that token position (left), and increase in probability for $c_2$ at the following token position. See Figure 2 and Section 2 for details.
*Head 14.1 scores an order of magnitude higher than the second-highest scoring head on the right plot. Its concept copying score is 0.0011. For visibility, we scale instead by the second-highest concept copying score.*

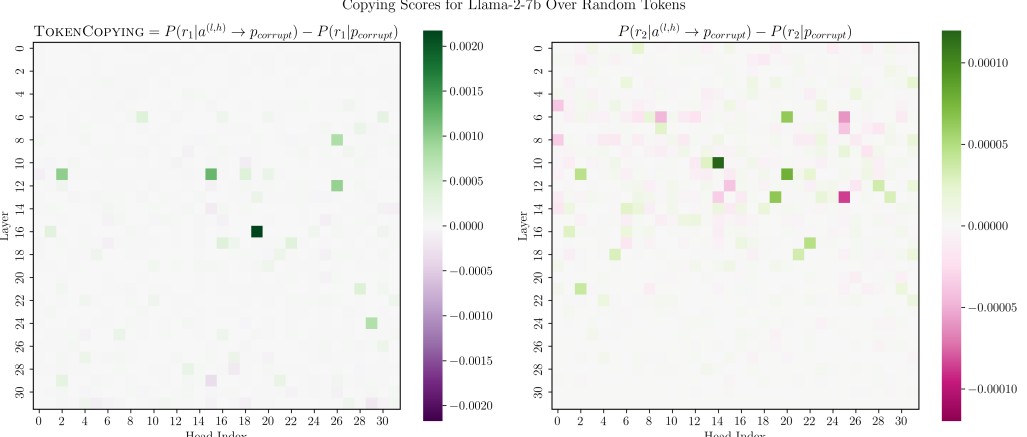

Figure 10: Probability differences when patching head activations for **Llama-2-7b** over **random tokens**. The left-hand side shows token copying scores; we do not utilize the right-hand scores in this work. We patch at token position $x'_n$ and measure increase in probability for $r_1$ at that token position (left), and increase in probability for $r_2$ at the following token position. See Figure 2 and Section 2 for details.

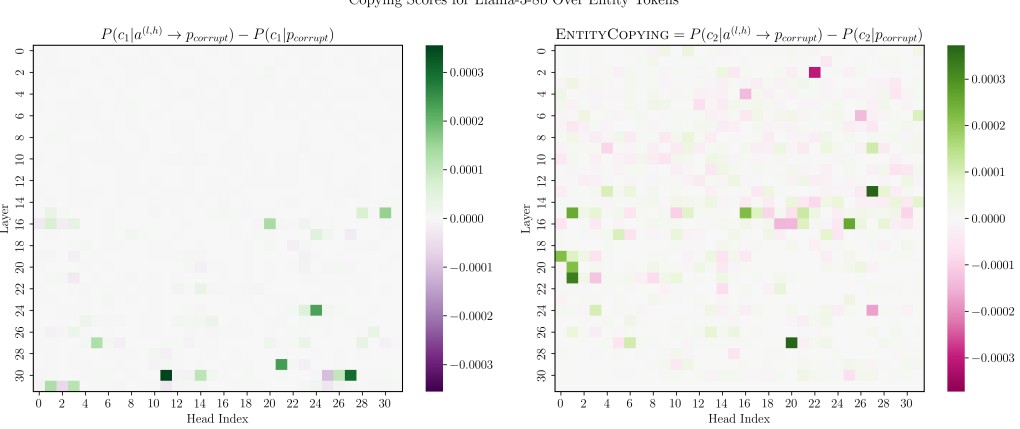

Figure 11: Probability differences when patching head activations for **Llama-3-8b** over **entity tokens**. The right-hand side shows concept copying scores; we do not utilize the left-hand scores in this work. We patch at token position $x'_n$ and measure increase in probability for $c_1$ at that token position (left), and increase in probability for $c_2$ at the following token position. See Figure 2 and Section 2 for details.

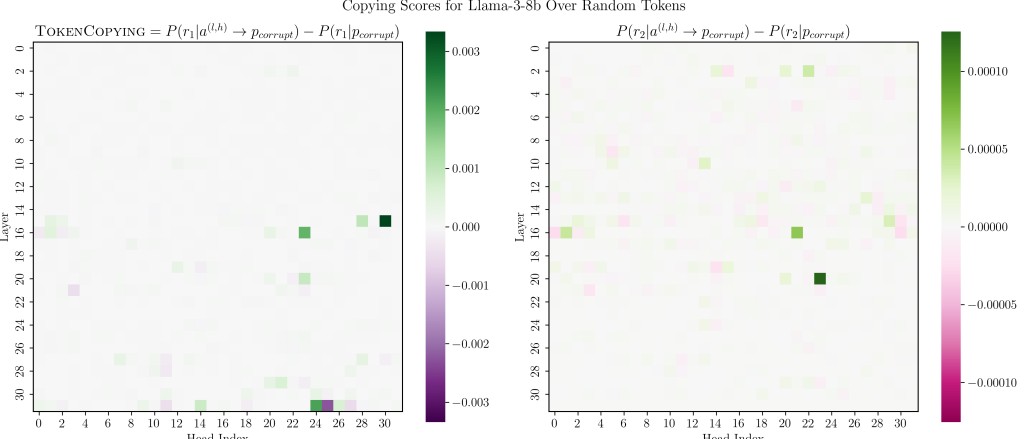

Figure 12: Probability differences when patching head activations for **Llama-3-8b** over **random tokens**. The left-hand side shows token copying scores; we do not utilize the right-hand scores in this work. We patch at token position $x'_n$ and measure increase in probability for $r_1$ at that token position (left), and increase in probability for $r_2$ at the following token position. See Figure 2 and Section 2 for details.

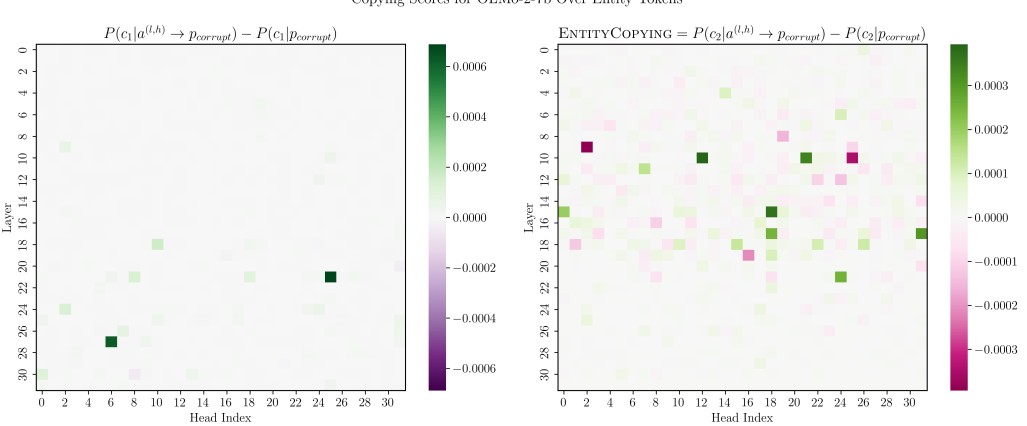

Figure 13: Probability differences when patching head activations for **OLMo-2-7b** over **entity tokens**. The right-hand side shows concept copying scores; we do not utilize the left-hand scores in this work. We patch at token position $x'_n$ and measure increase in probability for $c_1$ at that token position (left), and increase in probability for $c_2$ at the following token position. See Figure 2 and Section 2 for details.

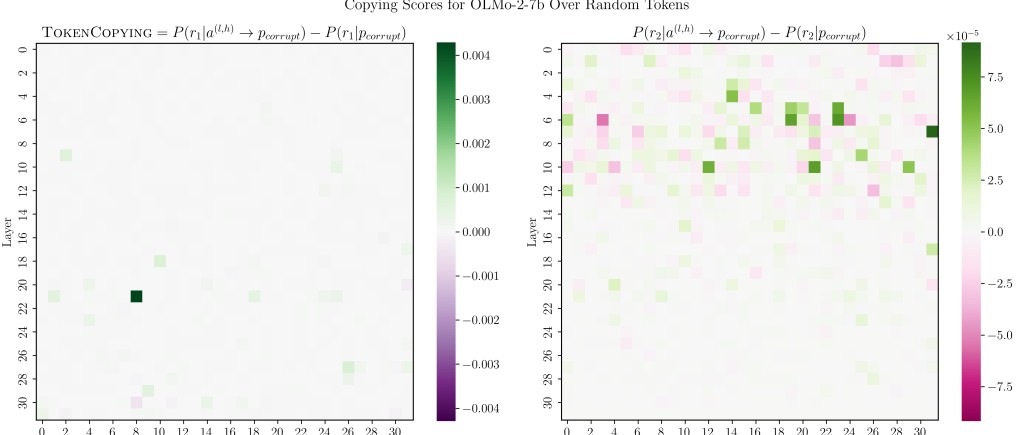

Figure 14: Probability differences when patching head activations for **OLMo-2-7b** over **random tokens**. The left-hand side shows token copying scores; we do not utilize the right-hand scores in this work. We patch at token position $x'_n$ and measure increase in probability for $r_1$ at that token position (left), and increase in probability for $r_2$ at the following token position. See Figure 2 and Section 2 for details.

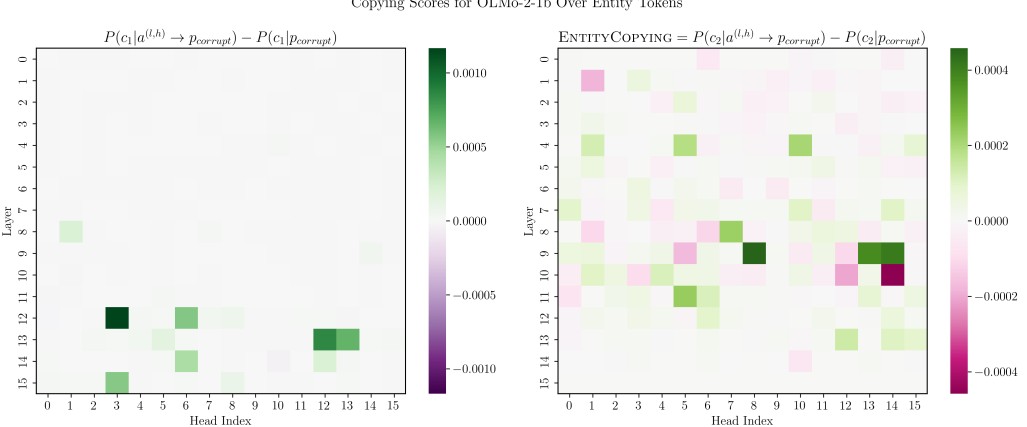

Figure 15: Probability differences when patching head activations for **OLMo-2-1b** over **entity tokens**. The right-hand side shows concept copying scores; we do not utilize the left-hand scores in this work. We patch at token position $x'_n$ and measure increase in probability for $c_1$ at that token position (left), and increase in probability for $c_2$ at the following token position. See Figure 2 and Section 2 for details. Unlike other models in this paper, OLMo-2-1b has only 16 layers and 16 heads per layer.

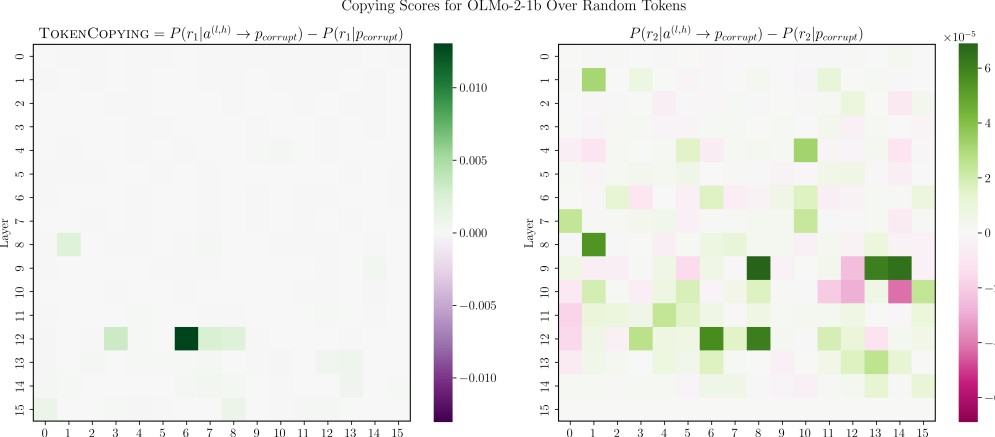

Figure 16: Probability differences when patching head activations for **OLMo-2-1b** over **random tokens**. The left-hand side shows token copying scores; we do not utilize the right-hand scores in this work. We patch at token position $x'_n$ and measure increase in probability for $r_1$ at that token position (left), and increase in probability for $r_2$ at the following token position. See Figure 2 and Section 2 for details. Unlike other models in this paper, OLMo-2-1b has only 16 layers and 16 heads per layer.

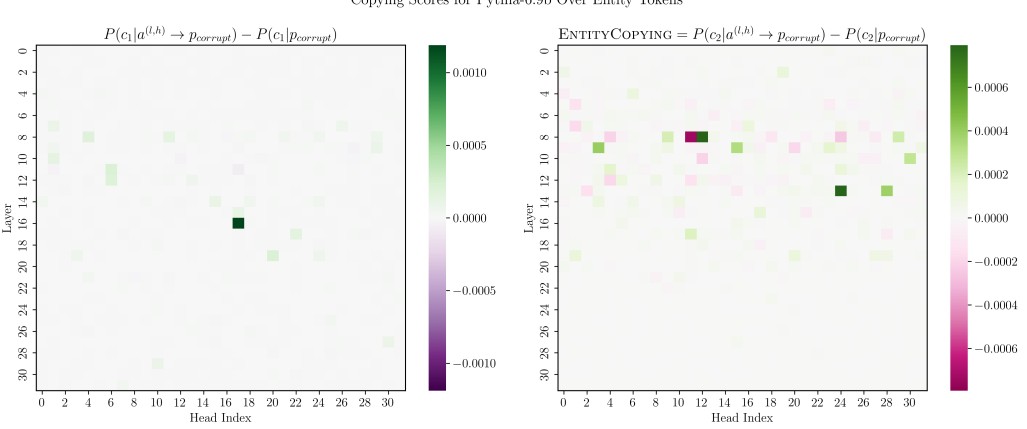

Figure 17: Probability differences when patching head activations for **Pythia-6.9b** over **entity tokens**. The right-hand side shows concept copying scores; we do not utilize the left-hand scores in this work. We patch at token position $x'_n$ and measure increase in probability for $c_1$ at that token position (left), and increase in probability for $c_2$ at the following token position. See Figure 2 and Section 2 for details.

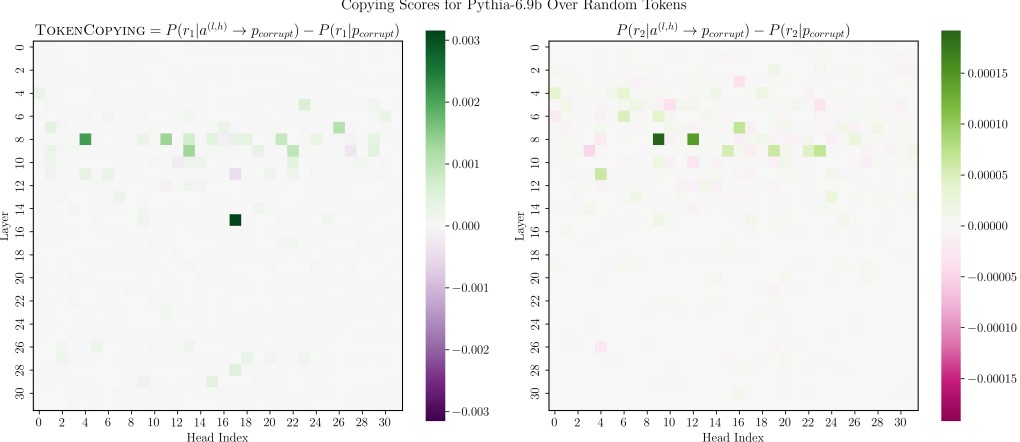

Figure 18: Probability differences when patching head activations for **Pythia-6.9b** over **random tokens**. The left-hand side shows token copying scores; we do not utilize the right-hand scores in this work. We patch at token position $x'_n$ and measure increase in probability for $r_1$ at that token position (left), and increase in probability for $r_2$ at the following token position. See Figure 2 and Section 2 for details.

## B  Token and Concept Induction Heads Throughout Training

We include analysis of causal and attention-based induction scores over time for OLMo-2-7b and Pythia-6.9b (models that provide access to training checkpoints). In Figures 19 and 21, we show token and concept copying scores from Section 2 over checkpoints. Figures 20 and 22 show next-token and last-token attention scores from Section 3 over checkpoints.

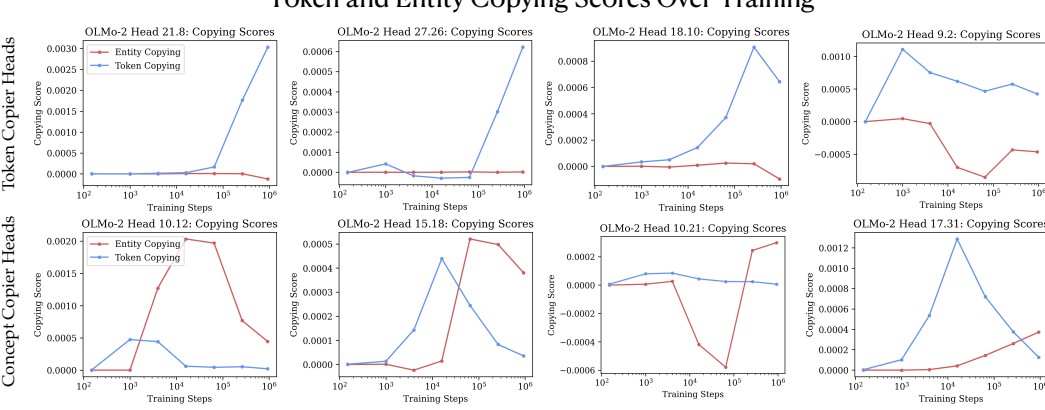

Figure 19: Causal copying scores from Section 2 for **OLMo-2-7b** throughout training checkpoints. Top row: top-4 token induction heads. Bottom row: top-4 concept induction heads.

Next- and Last-Token Matching Scores Over Training

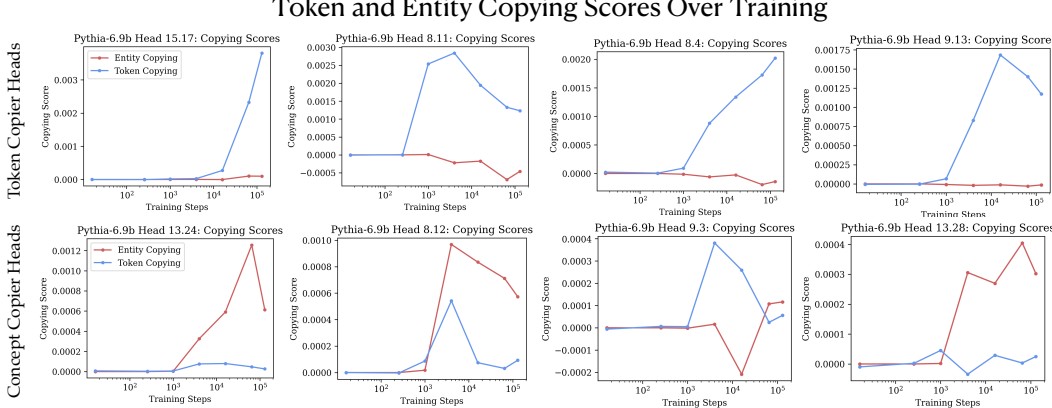

Figure 20: Attention-based matching scores from Section 3 for **OLMo-2-7b** throughout training checkpoints. Top row: top-4 token induction heads. Bottom row: top-4 concept induction heads.

Token and Entity Copying Scores Over Training

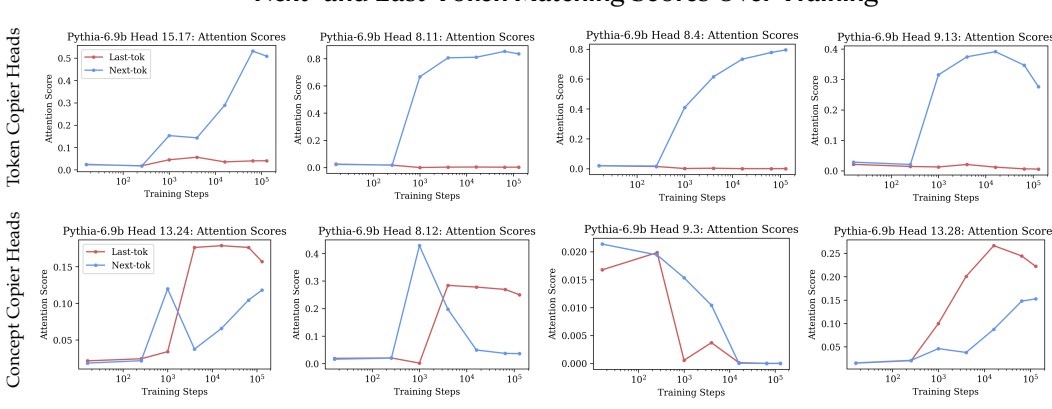

Figure 21: Causal copying scores from Section 2 for **Pythia-6.9b** throughout training checkpoints. Top row: top-4 token induction heads. Bottom row: top-4 concept induction heads.

Next- and Last-Token Matching Scores Over Training

Figure 22: Attention-based matching scores from Section 3 for **Pythia-6.9b** throughout training checkpoints. Top row: top-4 token induction heads. Bottom row: top-4 concept induction heads.

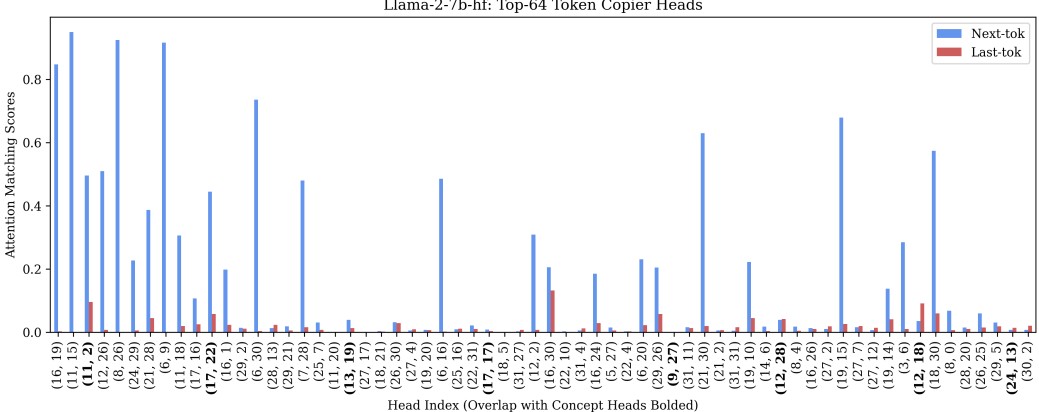

Figure 23: Next-token and last-token matching scores from Section 3 for the top-64 **Llama-2-7b token copier heads**. Heads that are also in the top-64 concept heads are bolded.

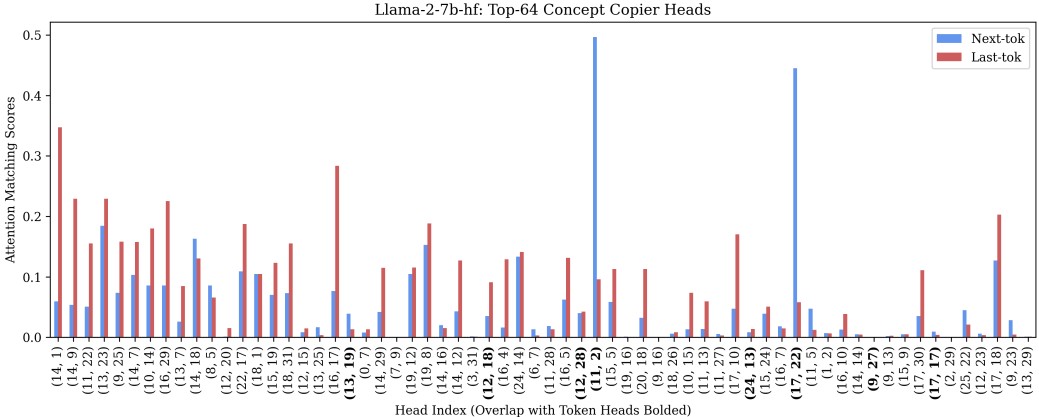

Figure 24: Next-token and last-token matching scores from Section 3 for the top-64 **Llama-2-7b concept copier heads**. Heads that are also in the top-64 token heads are bolded.

## C   Next-Token and Last-Token Matching Scores

### C.1   Attention Scores for Top 64 Heads

We include expanded versions of Figure 4 for the top-64 token and concept copier heads for each model. See Figures 23 through 30. We see that concept copier heads tend to have high last-token matching scores (§3), but also that each model has a few heads that seem to do *both* token and concept induction. Heads that appear in the top-64 of both token-copying scores and concept-copying scores are bolded.

### C.2   Correlations With Causal Scores

Figure 31 shows correlations between causal scores (Section 2) and attention-based scores (Section 3). For token induction, we compare token copying scores with next-token matching scores. For concept induction, we compare concept copying scores with last-token matching scores. Although these scatterplots appear quite noisy, there does seem to be a significant correlation between attention paid to the next/last token and propensity to copy tokens/entities respectively. Correlations for concept induction are comparable to correlations for token induction heads (which have been established in prior work).

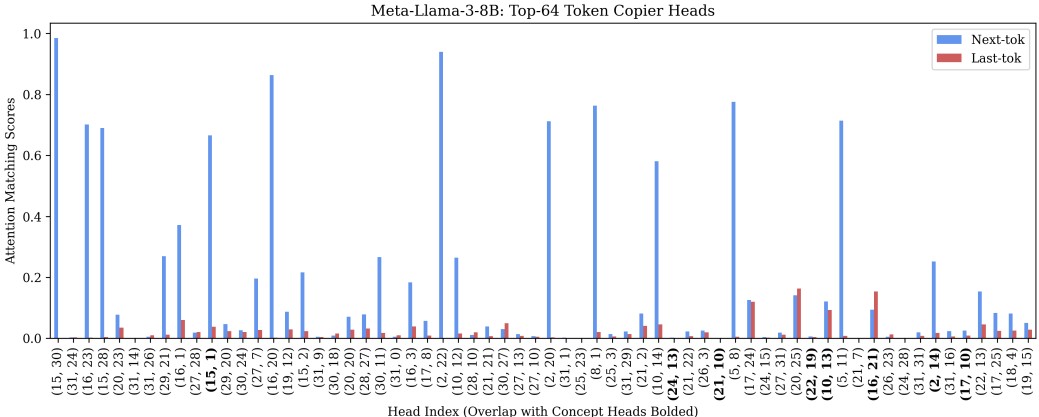

Figure 25: Next-token and last-token matching scores from Section 3 for the top-64 **Llama-3-8b token copier heads**. Heads that are also in the top-64 concept heads are bolded.

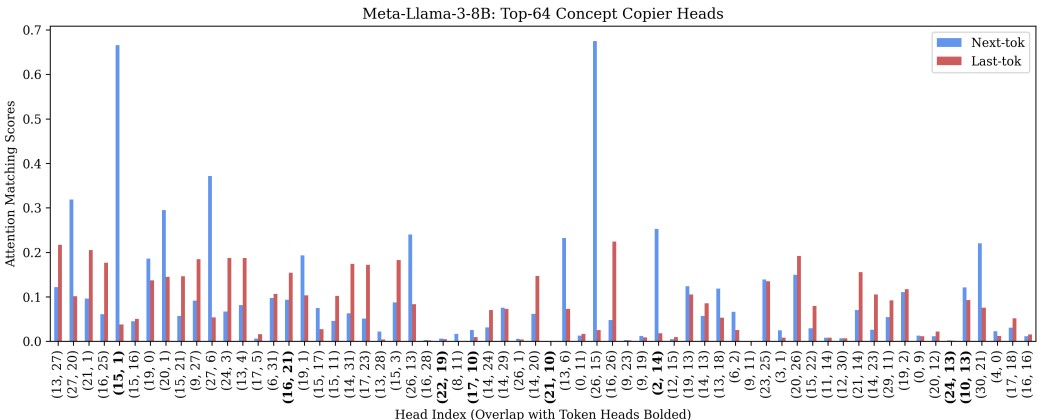

Figure 26: Next-token and last-token matching scores from Section 3 for the top-64 **Llama-3-8b concept copier heads**. Heads that are also in the top-64 token heads are bolded.

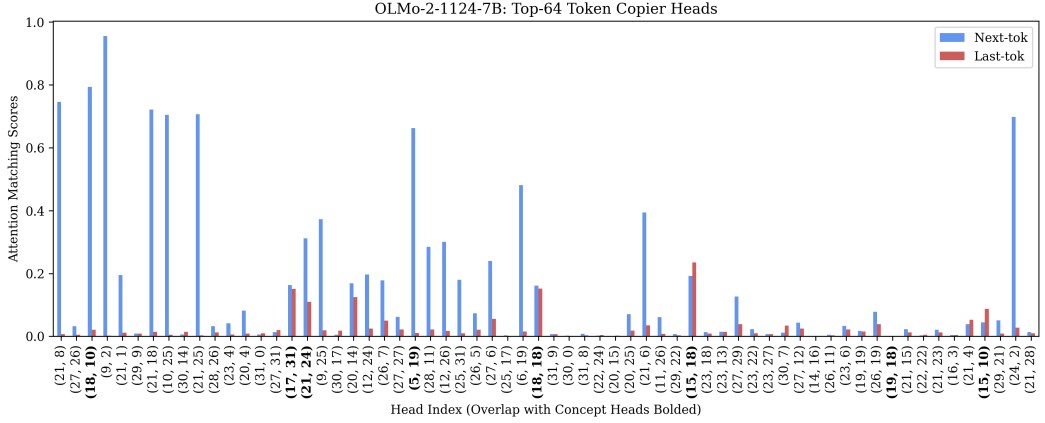

Figure 27: Next-token and last-token matching scores from Section 3 for the top-64 **OLMo-2-7b token copier heads**. Heads that are also in the top-64 concept heads are bolded.

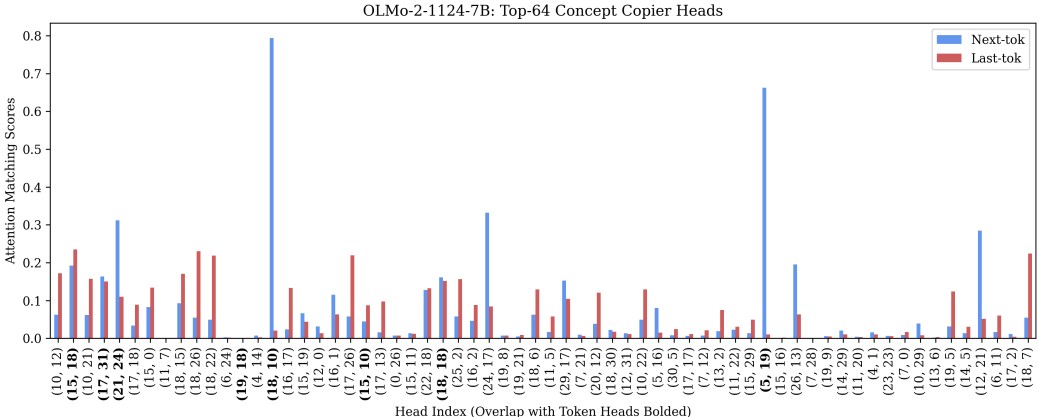

Figure 28: Next-token and last-token matching scores from Section 3 for the top-64 **OLMo-2-7b concept copier heads**. Heads that are also in the top-64 token heads are bolded.

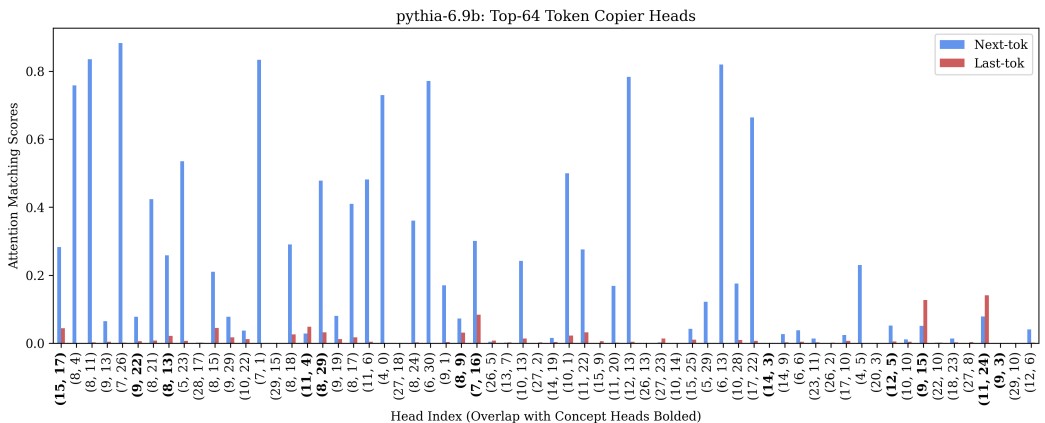

Figure 29: Next-token and last-token matching scores from Section 3 for the top-64 **Pythia-6.9b token copier heads**. Heads that are also in the top-64 concept heads are bolded.

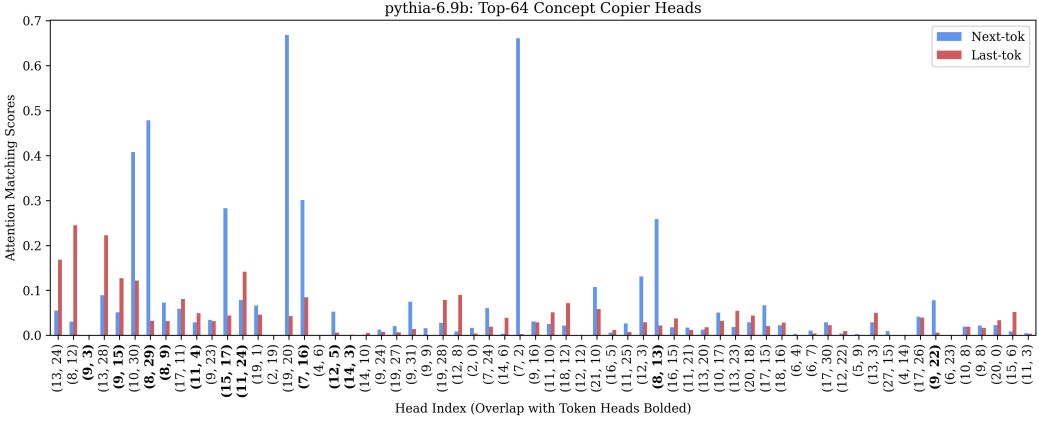

Figure 30: Next-token and last-token matching scores from Section 3 for the top-64 **Pythia-6.9b concept copier heads**. Heads that are also in the top-64 token heads are bolded.

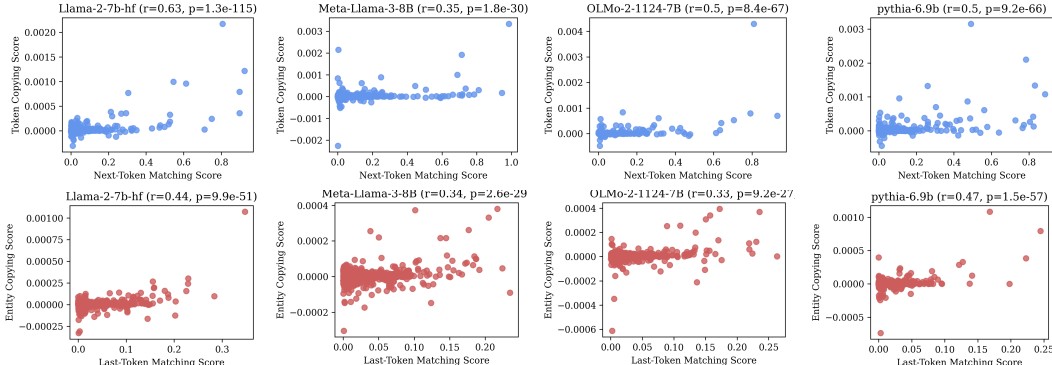

Figure 31: Relationship between attention-based matching scores (Section 3) and causal copying scores (Section 2). Top: correlations between next-token matching scores and token copying scores for each head, indicators of token-level induction. Bottom: correlations between last-token matching scores and concept copying scores for each head, indicating concept induction.

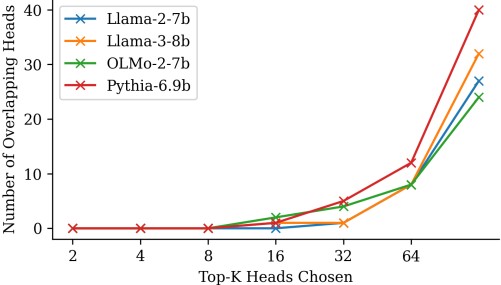

Figure 32: Number of overlapping token and concept copying heads for increasing values of $k$. For smaller values of $k$, OLMo-2-7b and Pythia-6.9b have more overlapping heads than Llama models.

# D  Lesioning Concept and Token Copier Heads

## D.1  Full Results

We show the results of ablating token copier heads and concept copier heads on our "vocabulary list" tasks for all four models in Figures 33-37. We also show the number of heads that overlap for increasing choices of $k$ in Figure 32.

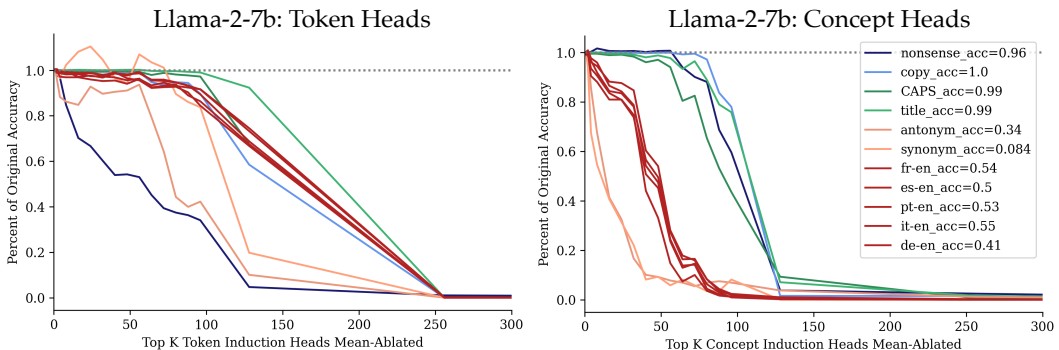

Figure 33: Mean-ablating the top-$k$ copier heads for "vocabulary list" tasks in **Llama-2-7b**. As results are shown as a percentage of original task accuracy, we display these full model accuracies in the legend. Even though synonym accuracy is initially low, it is still unaffected by ablation of token induction heads.

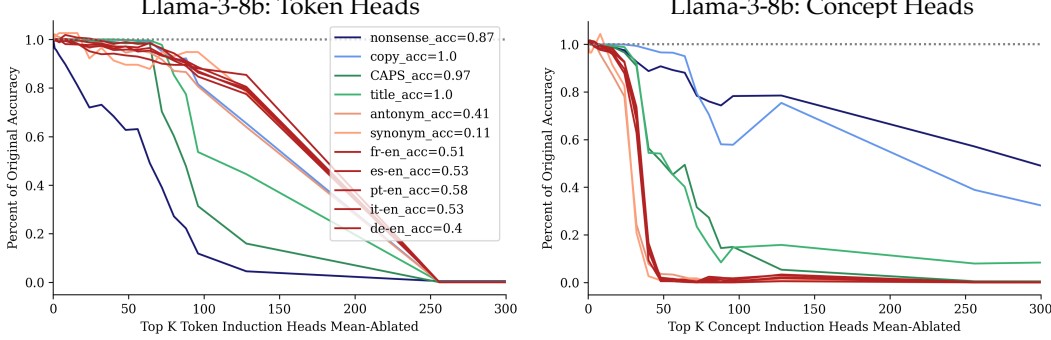

Figure 34: Mean-ablating the top-$k$ copier heads for "vocabulary list" tasks in **Llama-3-8b**. As results are shown as a percentage of original task accuracy, we display these full model accuracies in the legend. In contrast to Llama-2-7b, capitalization tasks seem to use a blend of concept induction and token induction heads.

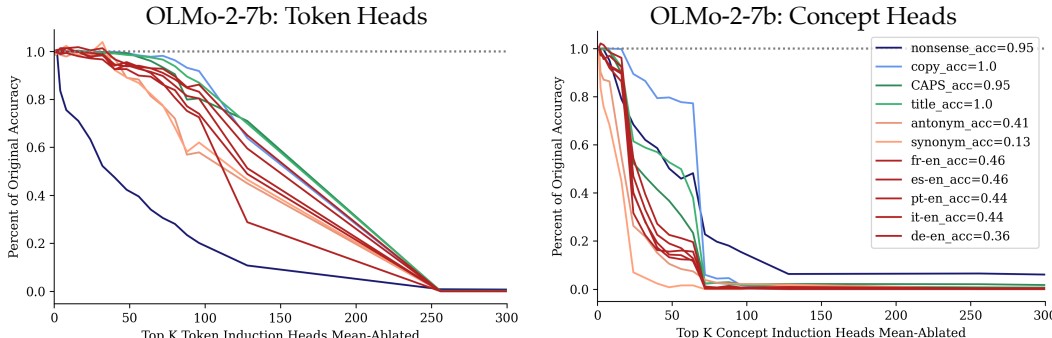

Figure 35: Mean-ablating the top-*k* copier heads for "vocabulary list" tasks in **OLMo-2-7b**. As results are shown as a percentage of original task accuracy, we display these full model accuracies in the legend. Although English copying accuracy remains high in the right figure, ablation of concept heads in OLMo-2-7b does damage nonsense token accuracy; this indicates that there may be more overlap between these heads for OLMo-2-7b than for Llama models.

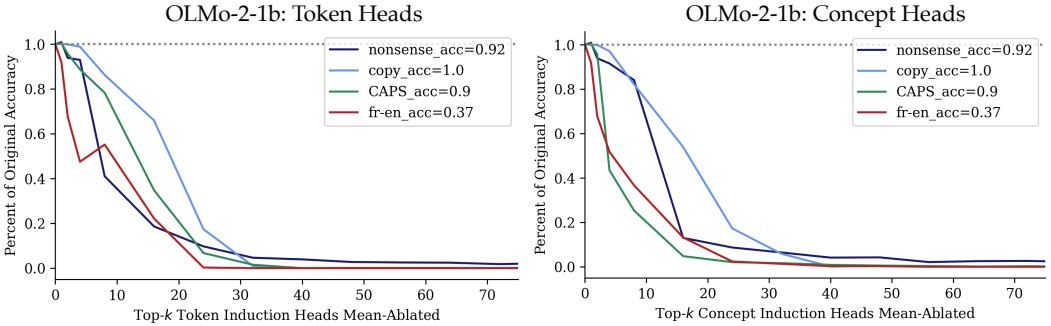

Figure 36: Mean-ablating the top-*k* copier heads for "vocabulary list" tasks in **OLMo-2-1b**. As results are shown as a percentage of original task accuracy, we display these full model accuracies in the legend. To be consistent with larger models, we show approximately the top 30% of heads.

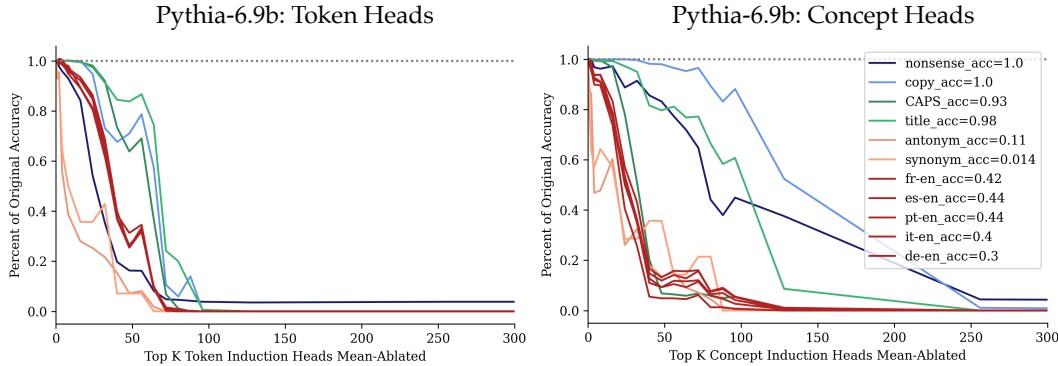

Figure 37: Mean-ablating the top-*k* copier heads for "vocabulary list" tasks in **Pythia-6.9b**. As results are shown as a percentage of original task accuracy, we display these full model accuracies in the legend. Unlike previous models, semantic tasks (translation, synonyms, and antonyms) are also damaged when we ablate token induction heads (left). However, capitalization and English copying accuracy remains relatively high. Because Pythia is trained on an order of magnitude fewer tokens than other models in this work (0.3 trillion, versus ≥ 2 trillion), it may be that concept induction and token induction are somewhat fused.

## D.2 Examples of Prompts

We show examples of two semantic task prompts from Section 4 in red boxes below. We also show examples of verbatim tasks in blue.

**Vocabulary List Prompt Example (French-English)**

```
 French vocab:
 1. completement
 2. inconstitutionnalité
 3. flocon
 4. chiens
 5. vaccinés
 6. racontée
 7. spécialités
 8. parfumée
 9. condensateur
 10. saumons
English translation:
 1. completely
 2. unconstitutionality
 3. flake
 4. dogs
 5. inoculated
 6. told
 7. specialties
 8. fragrant
 9. capacitor
 10.
```

**Vocabulary List Prompt Example (Antonym)**

```
 Words:
 1. hydrous
 2. enlarge
 3. alternative
 4. messy
 5. traditionalism
 6. simplicity
 7. underwater
 8. transitive
 9. omit
 10. contract
Antonyms:
 1. anhydrous
 2. cut
 3. primary
 4. neat
 5. rationalism
 6. difficulty
 7. amphibious
 8. intransitive
 9. remember
 10.
```

**Vocabulary List Prompt Example (English)**

```
 English vocab:
 1. live
 2. begin
 3. stumble
 4. good
 5. ostracize
 6. important
 7. coin
 8. colored
 9. manner
 10. recover
English vocab:
 1. live
 2. begin
 3. stumble
 4. good
 5. ostracize
 6. important
 7. coin
 8. colored
 9. manner
 10.
```

**Vocabulary List Prompt Example (Nonsense Copying)**

```
 Vocab:
 1. any insp
 2. comes look
 3. the like
 4.points_
 5.   fix
 6. $$_
 7.ence Camer
 8. there
 9. object currently
 10. ercase
Vocab:
 1. any insp
 2. comes look
 3. the like
 4.points_
 5.   fix
 6. $$_
 7.ence Camer
 8. there
 9. object currently
 10.
```

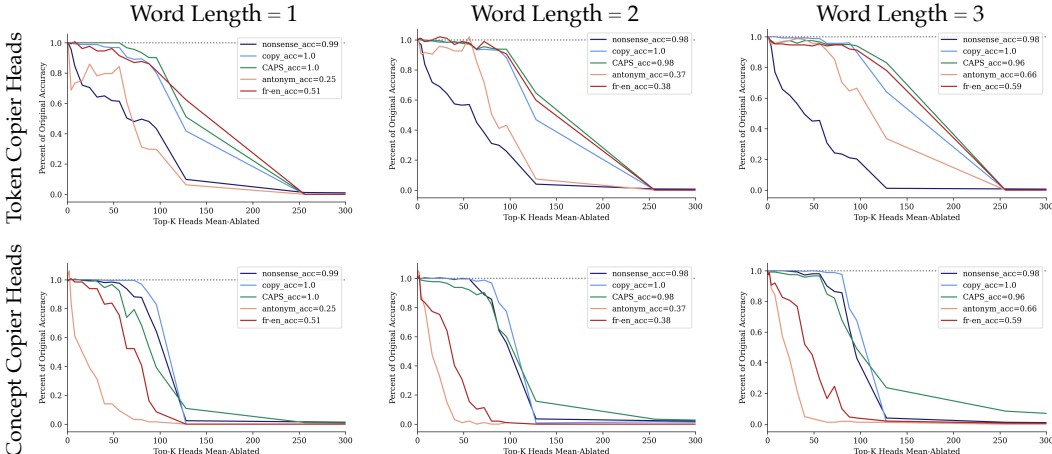

Figure 38: Ablations for Llama-2-7b, controlling the number of tokens in each word. When translating single-token French words to English, ablation of concept heads is less effective, suggesting that token heads may also do semantic copying when words are restricted to single tokens. However, the separation is still apparent, and is even clearer for two- and three-token words.

## D.3 Word Length Breakdown

We run the same ablation experiment as shown in Figure 5 for Llama-2-7b, except we control the number of tokens in each word. We run only on a subset of representative tasks and for $n = 256$ prompts per task. Figure 38 shows these results. While single-token words still have a similar pattern to multi-token words, the effect is less drastic, likely because the difference between concepts and tokens is less clear for single-token words.

## D.4 Qualitative Examples

We show examples of model generations with token induction heads ablated, expanding on examples shown in Section 4.2. We generate text with token induction heads mean-ablated at all token positions using the `nnsight.LanguageModel.generate` function (Fiotto-Kaufman et al., 2025), with default sampling. For the first three models, ablating $k = 32$ token induction heads causes the model to start paraphrasing instead of directly copying, for natural language (Boxes D.1, D.2, D.3, D.4) and code snippets (Boxes D.5, D.6, D.7).

We notice that for $k = 32$, Pythia-6.9b's "paraphrases" are much poorer than other models (Box D.3, D.7, D.11). We scale back to $k = 16$ and provide more than the first token of the copied sequence in Box D.4, finding that this causes Pythia-6.9b to copy more faithfully.

Interestingly, when paraphrasing a Python snippet, Llama-3-8b (Box 4.2) is the only model to write a snippet that is "correct," i.e. semantically equivalent to the original input.

We also show what model outputs look like for vocabulary list prompts: Llama-2-7b generates sentences that use previously-seen words (Box D.8), Llama-3-8b copies the list but without numbering (Box D.9), and OLMo-2-7b (Box D.10) and Pythia-6.9b (Box D.11) fail to copy any of the previously-seen words.

**Box D.1: (Llama-3-8b) Original Model vs. Top-32 Token Induction Heads Ablated**

I have reread, not without pleasure, my comments to his lines, and in many cases have caught myself borrowing a kind of opalescent light from my poet's fiery orb
I **have reread, not without pleasure, my comments to his lines, and in many cases have caught myself borrowing a kind of opalescent light from my poet's fiery orb**

- - - - - - - - - - - - - - - - - - - - - - - - - - - - - - - - - - - - - - - - - - - - - - -

...
I**'ve reread my comments on his lines, and found that I'd borrowed from his fiery orb a kind of opalescent light**

**Box D.2: (OLMo-2-7b) Original Model vs. Top-32 Token Induction Heads Ablated**

I have reread, not without pleasure, my comments to his lines, and in many cases have caught myself borrowing a kind of opalescent light from my poet's fiery orb
I **have reread, not without pleasure, my comments to his lines, and in many cases have caught myself borrowing a kind of opalescent light from my poet's fiery orb**

- - - - - - - - - - - - - - - - - - - - - - - - - - - - - - - - - - - - - - - - - - - - - - -

...
I **have read his lines again, and I have not found them without pleasure. In many cases, I have caught myself borrowing a kind of opalescent light from his fiery orb.**

**Box D.3: (Pythia-6.9b) Original Model vs. Top-32 Token Induction Heads Ablated**

I have reread, not without pleasure, my comments to his lines, and in many cases have caught myself borrowing a kind of opalescent light from my poet's fiery orb
I **have reread, not without pleasure, my comments to his lines, and in many cases have caught myself borrowing a kind of opalescent light from my poet's fiery orb**

- - - - - - - - - - - - - - - - - - - - - - - - - - - - - - - - - - - - - - - - - - - - - - -

...
I **have reread my comments on his lines, and I have found them to be, in general, true, and I have found myself to be in agreement with them.**

**Box D.4: (Pythia-6.9b*) Original vs. Top-16* Token Induction Heads Ablated**

I have reread, not without pleasure, my comments to his lines, and in many cases have caught myself borrowing a kind of opalescent light from my poet's fiery orb
I **have reread, not without pleasure, my comments to his lines, and in many cases have caught myself borrowing a kind of opalescent light from my poet's fiery orb**

- - - - - - - - - - - - - - - - - - - - - - - - - - - - - - - - - - - - - - - - - - - - - - -

...
I have reread, **not without pleasure, my comments to his lines, and have caught myself in many cases, borrowing a kind of opalescent light from my poet's fiery orb**

**Box D.5: (Llama-2-7b) Original Model vs. Top-32 Token Induction Heads Ablated**

```
foo = []
for i in range(len(bar)):
    if i % 2 == 0:
        foo.append(bar[i])
foo = []
for i in range(len(bar)):
    if i % 2 == 0:
        asdf.append(bar[i])
```

```
foo = []
for i in range(len(bar)):
    if i % 2 == 0:
        foo.append(bar[i])
foo = [item for sublist in
    bar for item in sublist
    if i % 2 == 0]
```

**Box D.6: (OLMo-2-7b) Original Model vs. Top-32 Token Induction Heads Ablated**

```
foo = []
for i in range(len(bar)):
    if i % 2 == 0:
        foo.append(bar[i])
foo = []
for i in range(len(bar)):
    if i % 2 == 0:
        asdf.append(bar[i])
```

```
foo = []
for i in range(len(bar)):
    if i % 2 == 0:
        foo.append(bar[i])
foo = [x for x in bar if
    x % 2 == 0]
```

**Box D.7: (Pythia-6.9b) Original Model vs. Top-32 Token Induction Heads Ablated**

```
foo = []
for i in range(len(bar)):
    if i % 2 == 0:
        foo.append(bar[i])
foo = []
for i in range(len(bar)):
    if i % 2 == 0:
        asdf.append(bar[i])
```

```
foo = []
for i in range(len(bar)):
    if i % 2 == 0:
        foo.append(bar[i])
foo = foo.append(foo)

print(foo)
```

**Box D.8: (Llama-2-7b) Original Model vs. Top-32 Token Induction Heads Ablated**

English vocab:
1. hospital
2. visage
3. unlucky
4. lesbian
5. know
6. wizard
7. unfair
8. sound
9. cortical
10. obsolete
English **vocab:**
**1. hospital**
**2. visage**
**3. unlucky**
**4. lesbian**
**5. know**
**6. wizard**
**7. unfair**
**8. sound**
**9. cortical**
**10. obsolete**

...
4. lesbian
5. know
6. wizard
7. unfair
8. sound
9. cortical
10. obsolete
English **vocabulary:**
**1. She was in hospital for a long time after the accident.**
**2. Her face was covered in bandages.**
**3. The accident was very unlucky.**
**4. She was a lesbian.**
**5. He knows everything about the wizard.**
**6. It is unfair to treat people like that.**
**7. The sound of the engine was very loud.**
**8. I don't know why the doctor told me that I should eat more fish.**
**9. The new cortical implants can help the blind to see.**
**10. The**

---

**Box D.9: (Llama-3-8b) Original Model vs. Top-32 Token Induction Heads Ablated**

English vocab:
1. hospital
2. visage
3. unlucky
4. lesbian
5. know
6. wizard
7. unfair
8. sound
9. cortical
10. obsolete
English **vocab:**
**1. hospital**
**2. visage**
**3. unlucky**
**4. lesbian**
**5. know**
**6. wizard**
**7. unfair**
**8. sound**
**9. cortical**
**10. obsolete**

English vocab:
1. hospital
2. visage
3. unlucky
4. lesbian
5. know
6. wizard
7. unfair
8. sound
9. cortical
10. obsolete
English **vocab:**
**hospital**
**visage**
**unlucky**
**lesbian**
**know**
**wizard**
**unfair**
**sound**
**cortical**
**obsolete**

---

**Box D.10: (OLMo-2-7b) Original Model vs. Top-32 Token Induction Heads Ablated**

English vocab:
1. hospital
2. visage
3. unlucky
4. lesbian
5. know
6. wizard
7. unfair
8. sound
9. cortical
10. obsolete
English **vocab:**
**1. hospital**
**2. visage**
**3. unlucky**
**4. lesbian**
**5. know**
**6. wizard**
**7. unfair**
**8. sound**
**9. cortical**
**10. obsolete**

English vocab:
1. hospital
2. visage
3. unlucky
4. lesbian
5. know
6. wizard
7. unfair
8. sound
9. cortical
10. obsolete
English **to French**
**French to English**
**English to French**
**French to English**
**German to English**
**Russian to English**
**German to Russian**
**English to German<|endoftext|>**

**Box D.11: (Pythia-6.9b) Original Model vs. Top-32 Token Induction Heads Ablated**

English vocab:
1. hospital
2. visage
3. unlucky
4. lesbian
5. know
6. wizard
7. unfair
8. sound
9. cortical
10. obsolete
English **vocab:**
**1. hospital**
**2. visage**
**3. unlucky**
**4. lesbian**
**5. know**
**6. wizard**
**7. unfair**
**8. sound**
**9. cortical**
**10. obsolete**

English vocab:
1. hospital
2. visage
3. unlucky
4. lesbian
5. know
6. wizard
7. unfair
8. sound
9. cortical
10. obsolete
English **vocabularies**
**\n English-French**
**\n English-French dictionary**
**\n English-French dictionary**
**\n English-French dictionary**
**\n English-French dictionary**
**\n English-French dictionary**
**\n English-French dictionary**
**\n English-French dictionary**
**\n English-French dictionary**

# E   Concept and Token Lens

## E.1   Approach

Every attention head in an autoregressive transformer can be thought of as "reading" some small amount of information from all previous token positions and subsequently "writing" that information back into the current residual stream. Can we visualize the semantic information that concept induction heads are contributing to the residual stream?

Let $d$ be a model's hidden dimension and $m < d$ be the dimension of a single head. We rely on a key insight from Elhage et al. (2021): that the value and output projections for a particular head $h$ at layer $l$, $V_{(l,h)} \in \mathbb{R}^{(m,d)}$ and $O_{(l,h)} \in \mathbb{R}^{(d,m)}$ respectively, are solely responsible for whatever information a head writes into the residual stream. Specifically, they point out that the product of these two matrices $O_{(l,h)} V_{(l,h)}$ is a low-rank $d \times d$ matrix (at most rank $m$) that determines the effect of head $(l,h)$ on the residual stream. In other words, multiplying a hidden state $x_l$ by this matrix extracts whatever information within $x_l$ that this head typically contributes to the residual stream.

To build a *concept lens* $L_{C_k} \in \mathbb{R}^{(d,d)}$ that reads from all of the concept induction head subspaces simultaneously, we combine the weights from the top-$k$ concept induction heads $C_k$. We choose $k = 80$ based on results from Section 6, and calculate the sum of all concept OV matrices:

$$L_{C_k} = \sum_{(l,h) \in C_k} V_{(l,h)} O_{(l,h)}. \tag{5}$$

If all attention heads in $C_k$ are in the same layer, $L_{C_k} x_l$ is mathematically equivalent to taking the sum of the outputs of those attention heads. However, we also allow for summation of heads across layers, which was empirically effective in prior work (Todd et al., 2024), possibly because transformer representations are interchangeable in intermediate layers (Lad et al., 2025).

Finally, we can project $L_{C_k} x_l$ to token space by applying the model's final normalization module and decoder head (nostalgebraist, 2020). This approach works for any set of heads: we can create a *token lens* $L_{T_k} \in \mathbb{R}^{(d,d)}$ out of the top-$k$ token induction heads, or a baseline lens $L \in \mathbb{R}^{(d,d)}$ as the sum of all OV matrices in the model.

## E.2   Lens Output Examples

Figures 39 through 42 show concept lens outputs for three different models. Compared to a baseline lens that sums all OV matrices (Figure 44), these plots reveal differing semantics of the same token in three different contexts. We also show examples of *token lens* in Figure 43, which appears ineffective for `inals` (the last token of a multi-token word) but clearly reveals token-level information at other positions. We posit that this discrepancy is a consequence of the "token erasure" effect found by Feucht et al. (2024).

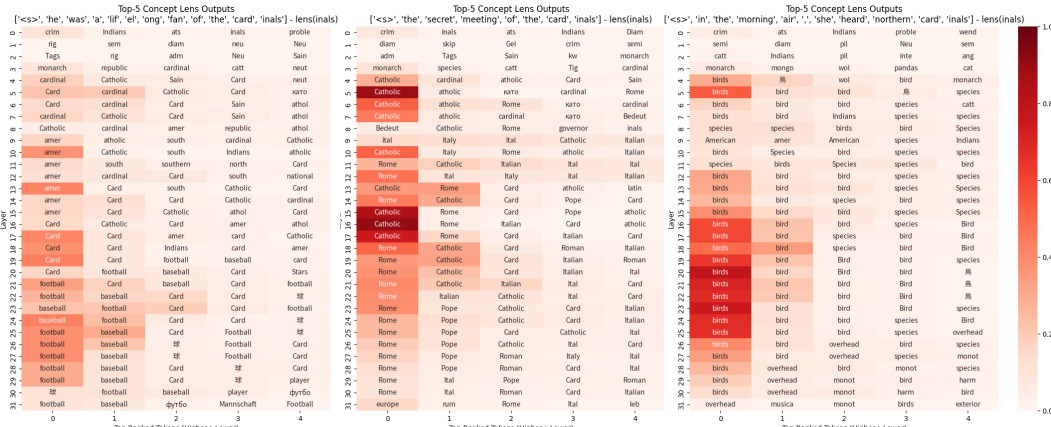

Figure 39: Concept lens outputs for **Llama-2-7b**. We multiply the hidden state for `inals` at every layer by $L_{C_k}$ (Equation 5) before projecting to vocabulary space. Applying this lens reveals the semantics of the token `inals`, which depends on the context.

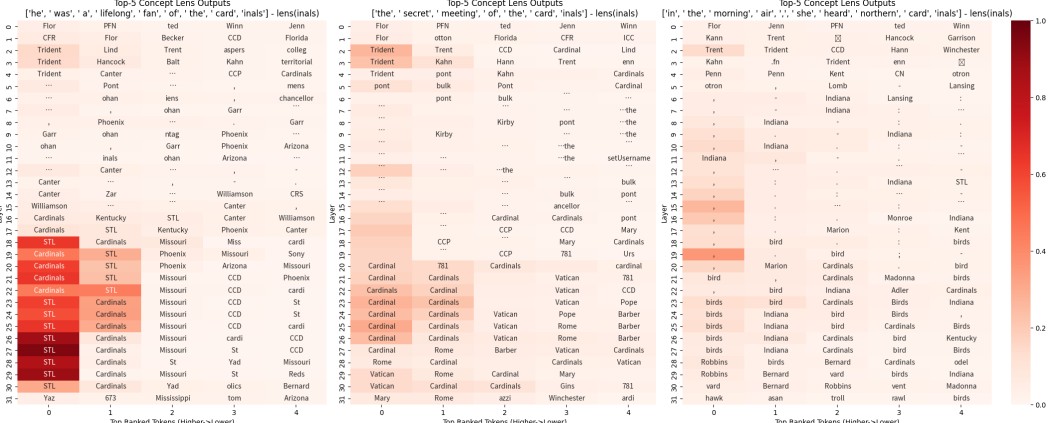

Figure 40: Concept lens outputs for **Llama-3-8b**. We multiply the hidden state for `inals` at every layer by $L_{C_k}$ (Equation 5) before projecting to vocabulary space. Applying this lens reveals the semantics of the token `inals`, which depends on the context. Unlike other models, early-middle layers are less decodable than middle-late layers. `STL` is likely a reference to the St. Louis Cardinals.

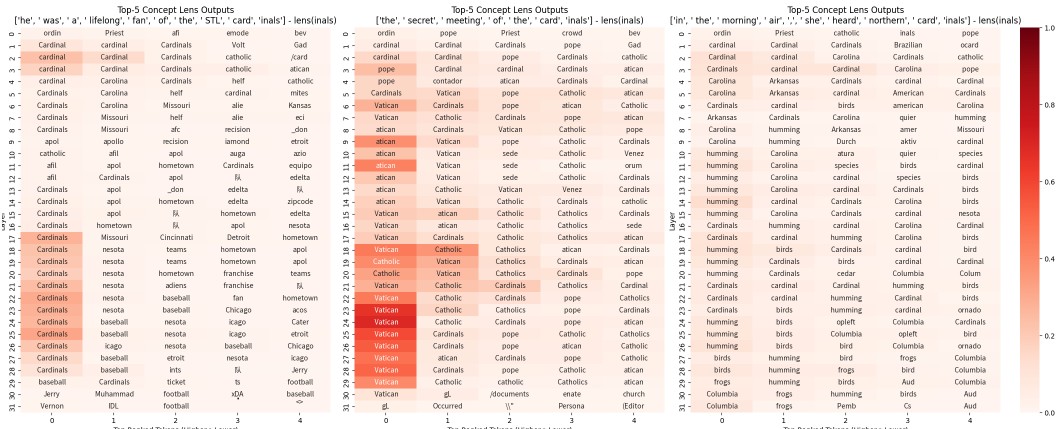

Figure 41: Concept lens outputs for **OLMo-2-7b**. We multiply the hidden state for `inals` at every layer by $L_{C_k}$ (Equation 5) before projecting to vocabulary space. Applying this lens reveals the semantics of the token `inals`, which depends on the context. Unlike Llama models, we must add "STL" to the leftmost prompt to decode semantics related to sports.

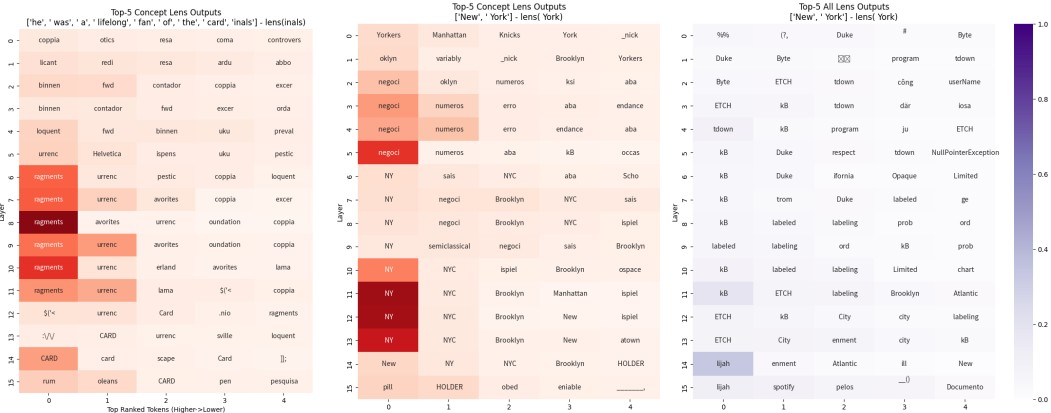

Figure 42: Concept lens outputs for **OLMo-2-1b**. We choose $k = 20$, i.e. the top 8% of heads, to be consistent with larger models. Unlike larger models, the signal provided by concept lens is quite noisy. However, for a simple multi-token concept like "New York", concept lens still reveals more semantic information than a baseline sum of all OV matrices.

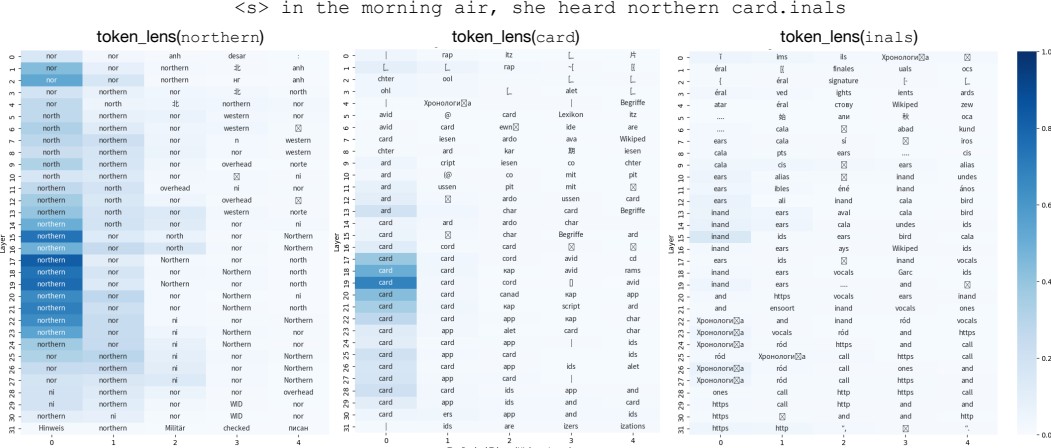

Figure 43: Token lens outputs for **Llama-2-7b** at three token positions. Applying token lens reveals the token that corresponds to a given hidden state. We multiply the hidden state for `inals` at every layer by $L_{T_k}$ before projecting to vocabulary space, with $k = 80$. Token lens is not effective for (`inals`), likely due to the "token erasure" effect for multi-token words described by Feucht et al. (2024).

Figure 44: Baseline "all heads" lens outputs for **Llama-2-7b** across three prompts. We multiply the hidden state for `inals` at every layer by $L$, the sum of all OV matrices in the model, before projecting to vocabulary space. Although we can still observe some semantic information when using all heads, concept lens (Figure 39) provides a cleaner signal.

## F  Concept Induction is Language-Agnostic

We show results from Section 6 for more languages, with both Llama models. Figure 45 shows results for the same experiment as in Figure 6, expanding evaluation to Llama-3-8b and two more language sets: patching from French-English into German-Russian, and patching from Russian-Spanish into English-Japanese. Figure 46 shows results for a smaller model from a different family, OLMo-2-1b.

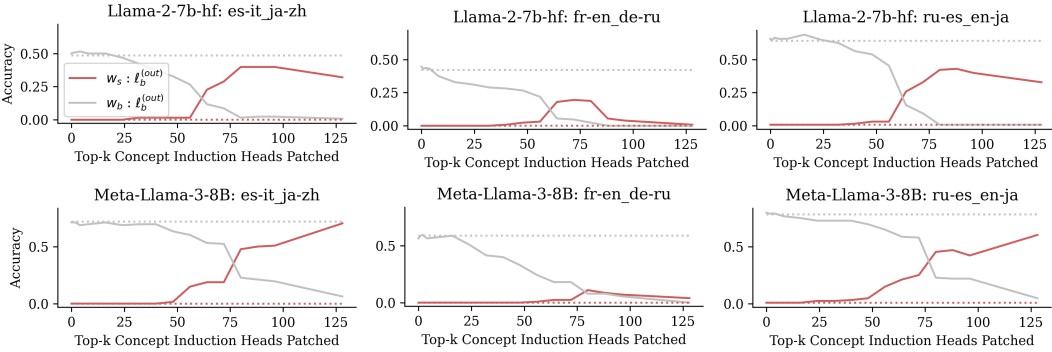

Figure 45: Results for patching the top-*k* concept induction heads from one translation prompt to another. Top-left is the same plot as Figure 6. We evaluate Llama-2-7b and Llama-3-8b on Spanish-Italian → Japanese-Chinese, French-English → German-Russian, and Russian-Spanish → English-Japanese. Interestingly, the language set for which concept patching is the *least* effective is the same language set for which FV head patching is *most* effective.

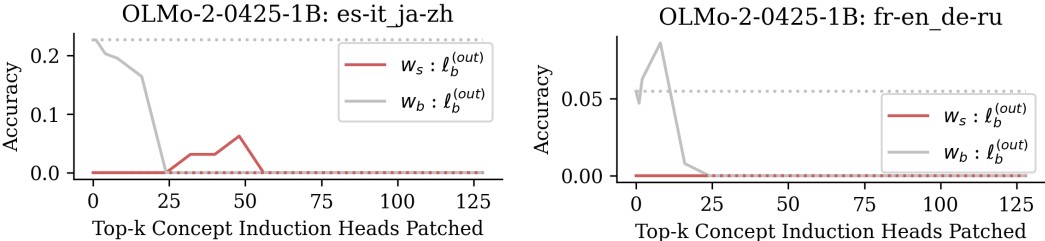

Figure 46: We also include results for OLMo-2-1b, patching from Spanish-Italian → Japanese-Chinese and French-English → German-Russian. The effect of patching concept heads is less clean than for larger models. This may be due to decreased separation between token and concept induction, or because of OLMo-2-1b has lower overall translation performance. Like larger Llama models, the latter language pair (fr-en→de-ru) shows weak results. Gray dotted lines indicate base model accuracy for Japanese-Chinese and German-Russian translation respectively.

## G    Function Vector Versus Concept Induction

As mentioned in Section 7, we find weak correlations between FV and concept copying scores (Figure 48). We also find that ablation of FV heads also causes a large drop in performance for vocabulary list tasks that cannot otherwise rely on token induction heads (Figure 49). However, this does not necessarily mean that FV heads play the same role as concept induction heads. As Section 7 explains, patching FV heads and patching concept induction heads for the same prompt yields very different results. While patching FV heads changes the *language* that the model outputs (depending on the language pair), patching concept induction heads changes the *meaning* of the output word. In order for the model to do semantic copying tasks, it needs both FV heads and concept heads—therefore, if either are ablated, we see the same drop in translation accuracy. We show results for patching FV heads for Llama models for two language sets in Figure 47.

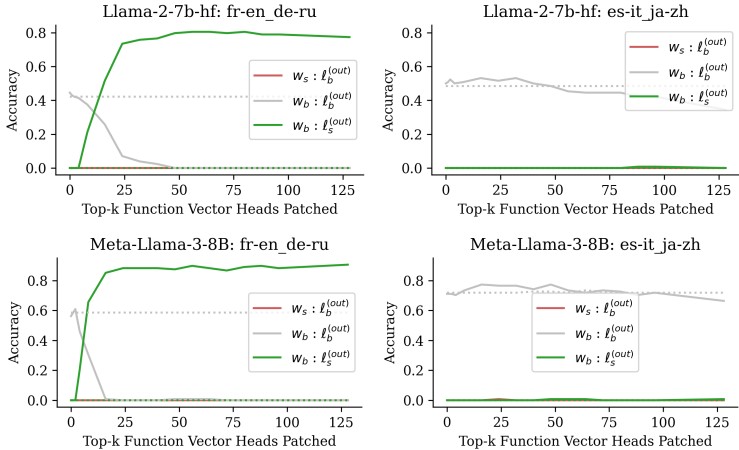

Figure 47: Results for patching the top-*k* FV heads from one translation prompt to another. Top-left is the same plot as Figure 7. Patching FV heads from Spanish-Italian to Japanese-Chinese does not flip the output language to Italian, but patching from French-English to German-Russian flips the output language to English.

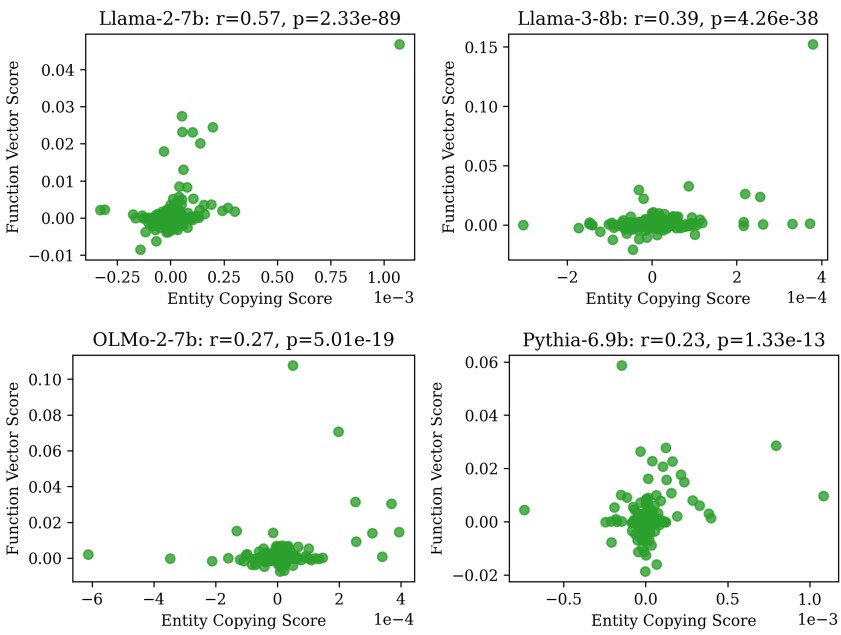

Figure 48: Correlations between FV scores and entity copying scores for all models. While we do find significant correlations, they are strongest for Llama models with strong outliers, and do not seem particularly strong for other heads and models.

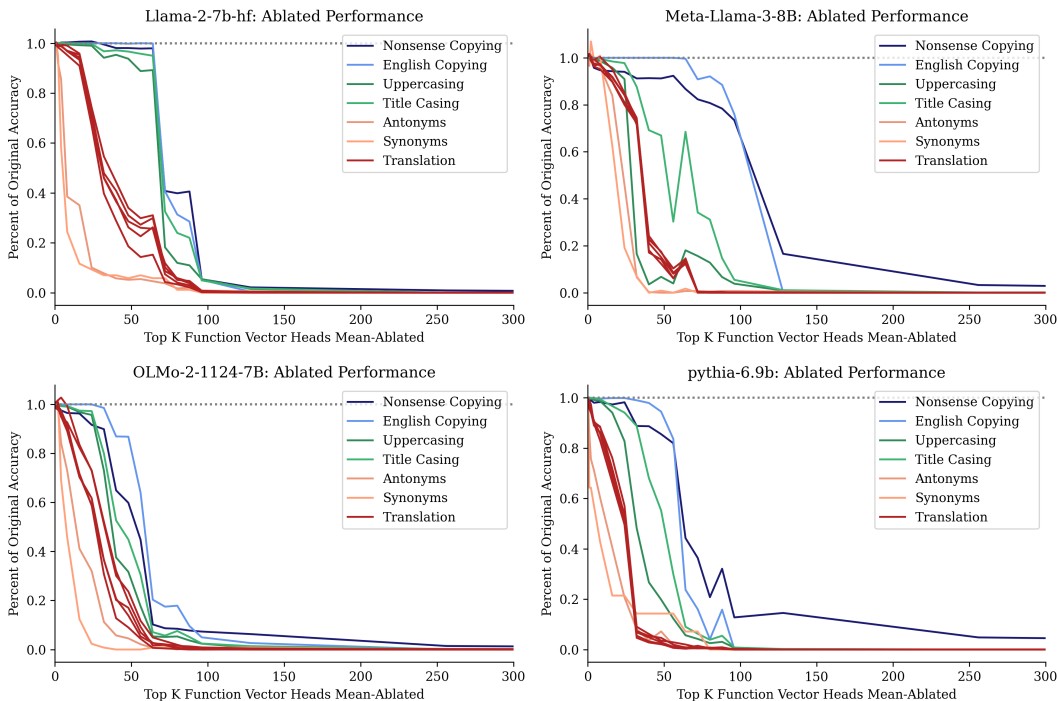

Figure 49: Ablation experiment described in Section 4 where we ablate the top-*k* function vector (FV) heads. We see that FV heads are also vital for semantic copying tasks. As Section 7 describes, this result is likely not due to an overlap with concept induction heads, but rather because FV heads help the model output the correct *language* (whereas concept heads copy the word *meaning*).

