# OpenReview forum: "The Dual-Route Model of Induction"
_colmweb.org/COLM/2025/Conference — COLM 2025_

### Official Review · Reviewer_NwaB · 2025-05-12

**Rating:** 5
**Confidence:** 3
**Ethics Flag:** 1

**Summary:**

This paper proposes a dual-route model of induction in LLMs, which consists of two distinct types of attention heads: (1) token induction heads that copy tokens verbatim, and (2) concept induction heads that copy entire lexical units. Using causal interventions and ablation studies across several models, the authors argue these two mechanisms support different in-context tasks, i.e., verbatim copying vs. semantic copying (e.g., translation). The work draws an analogy to human reading systems and explores language-agnostic representations via activation patching.

**Reasons To Accept:**

This paper introduces "concept induction heads" that enriches the taxonomy of known transformer circuits. The paper also presents analysis combining causal tracing, attention analysis, and ablations across multiple models and tasks. The paper can potentially offer a novel perspective of understanding in-context mechanisms.

**Reasons To Reject:**

- The paper is dense and difficult to follow, especially in key methodological sections (e.g., causal interventions, score definitions).

- The evaluations rely heavily on synthetic or narrow tasks. There's little evidence the phenomenon generalizes to real-world in-context learning.

- It's also not evident how identifying concept heads helps with model design, prompting, or downstream tasks. The contribution feels more observational than actionable to me.

---

> ### Author Response · Authors · 2025-05-28
>
> Thank you for your review! We’d like to share a few responses to your concerns:
>
> 1. We appreciate the feedback on our methodological explanations. If there are specific lines or sections that you find particularly confusing, we are more than happy to try and make those explanations clearer.
>
> 2. Our evaluation tasks are no more synthetic than any other standard in-context learning (ICL) tasks. In fact, they are extremely similar to tasks from Brown et al. (2020), the paper that originally coined the term “in-context learning”. We believe that these tasks are interesting to study because it is not immediately obvious why LLMs would be able to complete such unnatural sequences in the first place. As our qualitative examples suggest (Box 4.1 and 4.2), study of these capabilities in a controlled environment may provide insights that are applicable in much broader contexts.
>
> 3. If our ultimate goal is to understand how LLMs are able to do downstream tasks like translation, summarization, or paraphrasing, then we have to start with simpler tasks like word-level translation, before moving onto more complicated problems, like understanding syntax and discourse-level features. Our paper provides a foundation for that future work, by identifying how LLMs handle word-level concepts in ICL settings.

---

> > ### Comment · Reviewer_NwaB · 2025-06-09
> >
> > Thank you for the clarification. It makes sense to begin from a foundational perspective, and I’ve updated my evaluation accordingly. That said, it would be even more compelling to see connections to downstream applications.

---

### Official Review · Reviewer_sPEp · 2025-05-12

**Rating:** 5
**Confidence:** 3
**Ethics Flag:** 1

**Summary:**

In this paper, concept-level induction heads which copy entire lexical units in LLMs is introduced as a new type of induction head, not individual tokens. It works in parallel with token-level induction heads to copy meaningful text. It was indicated that these heads are important for semantic copying tasks, and they are language-agnostic.

**Reasons To Accept:**

This paper indicated that patching concept induction heads is effective in various LLMs based on detail experimental results.
Moreover, it showed concept heads attend to the end of the next word. That is, concept heads can transfer entire concepts at once.

**Reasons To Reject:**

The effectiveness of the proposed method is not clear in semantic task for the following reasons.

1. In language-agnostic of section 5, the accuracy by patching concept heads is about 0.4. Moreover, in also Figure 36, the accuracies by patching concept heads are under 0.5 except Spanish-Italian -> Japanese-Chinese and Russian-Spanish -> English-Japanese. Therefore, it is not clear what these experimental results mean in word-level translation task. This paper needs to indicate the effectiveness of the proposed method through comparison experiments with other previous works.

2. The experimental results by patching function vector (FV) heads of section 6 indicated that patching FV heads obtains output the same concept in a different language. However, patching FV heads may be effective comparison with patching concept heads from the viewpoint of accuracy because of accuracies French-English -> German-Russian are over 0.8. That is, it is not clear what was obtained through the experimental results in section 6.

---

> ### Author Response · Authors · 2025-05-28
>
> Thank you for your review and helpful feedback! We appreciate the skepticism for our section 5 and 6 figures. Your confusion is valid, and perhaps comes from the fact that we did not properly explain our baselines—we will fix this in the final draft.
>
> 1. We agree that an accuracy of 0.40 may appear quite low on its own, until you consider the model’s original word-level translation accuracy, which we plot as a gray dotted line. We can see from this dotted line in Figure 6b that Llama-2-7b’s original Japanese-Chinese word translation accuracy was only around 0.48. Therefore, patching concept head outputs recovers about 83% of the model’s original translation accuracy. We’ll try to clarify this in the caption.
>
> 2. You’re right that accuracy for patching FV heads is much higher than concept head accuracy, but this may be due to a quirk in FV behavior for English translation, which we mention in Appendix F. It seems that models are very eager to output English as opposed to, e.g., Russian. Thus, when we patch FV head outputs containing “English” into a new context, the model is *very* likely to output the correct word in English, whereas when we patch FV head outputs containing, e.g., “Italian” into a new context, the model’s behavior does not seem to change. Because this paper is not about FVs, we do not dedicate much space to this discrepancy. Regardless, Figure 7 still shows that FV heads are “sisters” to concept heads: when patched, they influence language information, instead of semantic information. We will update the main text to more clearly address this concern.

---

> > ### Author Response · Authors · 2025-06-09
> >
> > We'd like to remind the reviewer that the rebuttal/discussion phase is coming to an end, and we have not yet received a response. Here are the changes we made to the captions of Figure 6 and 7:
> >
> > Figure 6 caption:
> > > (a) Patching the top-$k$ concept induction head outputs from a Spanish-Italian prompt into a Japanese-Chinese prompt. (b) Patching concept induction heads changes the concept output by the model without affecting the language (i.e., ***niño*, the Spanish word for "child,"** is translated into Chinese instead of Italian). This effect is strongest for k=80, which is also where the largest separation between semantic and literal copying performance is found in Figure 5c. **Across n=128 examples, this approach causes the model to output the source Spanish word in the base output language with an accuracy of about 0.40 (solid red line). This is comparable to the model's original Japanese-Chinese translation accuracy of 0.48 (dotted gray line).**
> >
> > Figure 7 caption:
> > > (a) Patching concept heads changes semantics without affecting language, while patching FV heads changes output language without affecting meaning. In red, we show the same experiment from Figure 6: here, patching concept heads causes the model to output the source concept "committee" in Russian. In green, we show that patching FV heads at the same position for the same prompt causes the model to output the base concept "toilet" in English.
> > (b) **Across n=128 examples, patching FV heads from a French-English prompt into a German-Russian prompt flips the output language to English with an accuracy of about 0.80 (green line). This is higher than the model's original German-Russian translation accuracy (gray dotted line, approximately 0.41), perhaps because translating into English is easier for Llama-2-7b than translating into Russian.**
> >
> > We would appreciate your feedback on whether these changes address your concerns, or if there is more we can do do address your points. Thank you so much!

---

### Official Review · Reviewer_ZnwZ · 2025-05-13

**Rating:** 9
**Confidence:** 3
**Ethics Flag:** 1

**Summary:**

This paper introduces the concept-level induction at a multi-subword token level. The paper first measures the causal interventions effects by measuring the differences between concept vs token-level copying behavior. Then the next part of paper compares the difference between the concept-level vs token-level attention activation with respect to previous or next token and they showed that the concept level activation heads attends to the previous token. Next using various word pairs and perturbations/lesioning the word pairs to show that different pertubations affects tokens vs concept-level induction differently. And after, the paper did an experiment to patch/transplant the concept induction head from a translation prompt to another translation prompt for a different language pair and showed that the translation in the transplanted induction head can correct predict next token in the different language pair, proving the concept-level induction head as language agnostic (but this is only effective up to the top-80 concepts). Last hypothesis that the paper tested was the idea of function vector vs concept-level heads, the function level heads were supposed to be task-specific, some inconclusive results here and there for this part of the paper but the authors have shown how FV heads and concept-level heads perform differently.

**Questions To Authors:**

Food for thought: When the feature vector head is actually just 1 token, e.g. in older BERT-style language modeling we'll use a specific special token to simplify the preprocessing and training, wouldn't the FV head be the same as concept head?

Please do also take a look at pointer-network style attention, although it's not directly relevant to decoder-only architecture, the idea is similar when the head is pointing to prior tokens. The pointer-style would be similar to token-level heads https://arxiv.org/pdf/1506.03134 and even older morpheme vs concept level representation https://nlp.stanford.edu/pubs/conll13_morpho.pdf (which is something between concept vs token level)

**Reasons To Accept:**

- Really good job done at breaking down the proposal into provable hypotheses and gradually showing the effectiveness of the concept-level head empirically in each hypothesis test
- A lot of effort was put into the ablation studies for each hypotheses (though not in the main paper, but it's a comprehensive appendix of results)
- The proposed concept-level head was shown to behave consistently across various LMs (Olmo, pythia, llama)

**Reasons To Reject:**

- Nothing, good paper, clearly written, will fight to have it accepted.

---

> ### Author Response · Authors · 2025-05-28
>
> Thank you for the positive feedback, we are glad to hear that you find our experiments convincing!
>
> Your connection to specialized tokens in BERT-style models is an interesting parallel. In autoregressive models, information seems to be “forced” to collect at the end of important units (i.e., the ends of multi-token words or task descriptions), which FV and concept heads later attend to. But in a BERT-style model, the “forcing” comes from the fact that information needs to end up in the CLS token. If the model is fine-tuned to do a specific task, you could possibly think of FV and concept heads as doing the same thing (or one category being subsumed into the other).
>
> We will definitely take a look at prior work on pointer-network style attention. The paper you shared on morpheme vs. concept-level representation is also interesting: our framing of subword tokens as disjoint units here is a bit of an oversimplification, which misses out on all the richness of morphemic representations. We think this is an interesting area for future work.

---

### Official Review · Reviewer_3CWa · 2025-05-13

**Rating:** 8
**Confidence:** 3
**Ethics Flag:** 1

**Summary:**

This paper explores LLM copying behavior, expanding on prior work on induction heads. This paper fits into the “careful and deep analysis” dimension for COLM contributions. While there may be some near-term technological impact from the results, the main contribution is a deeper understanding of components of LLMs and how they impact LLM output. The insights may also contribute to understanding of translation processes in LLMs and will likely also inspire additional research and evaluation across a wider range of languages in the future. The paper uses established approaches to measure effects of causal interventions and ablations to examine the behavior of LLM heads.

**Questions To Authors:**

Questions and Comments:

- Line 31: This should probably be “pommes de terre” (not “des”)
- Line 73-74: “resampling” – to clarify, this is a sampling as described in the footnote, not simply a shuffling?
- Line 74: Should $y_{n+l}$ be $y_{n+m}$ ?
- Perhaps move lines 87-92 earlier in this section, e.g., between 72 and 73? (Not completely necessary and may not flow well, but I found myself wanting to be sure I understood what kinds of concepts were going to be used around that earlier point.)
- Figure 3: Consider using different shapes for the Token/Concept (will be clearer to readers who print in grayscale).
- Line 125: I was slightly confused by the description “insert the concept on the left”
- Figure 4: The two colors are very similar when printed in grayscale.
- Figure 5: Nearly impossible to read in grayscale. Would be nice to use different line patterns to distinguish.
- Figure 6: I found the parenthetical in the caption somewhat confusing and had to reread the section of the text to be sure I understood it. Perhaps there is a way to rephrase. If there’s not an easy way to fit a gloss into the figure, it may help to give English glosses in the caption.
- Figure 7 appears quite a bit before Function Vectors are first discussed, so it may be helpful to either move that or expand the acronym the first time that it appears in the caption.
- Lines 258-260: “FV heads cause the model to output the same concept in a different language, whereas concept induction heads cause the model to output a different concept in the same language” – should “patching” appear somewhere in this sentence?
- Line 683, box D.1: Is there a reason (other than space-saving) that you didn’t split these into two separate figures? As a reader, my eyes went (very confusingly) from “1. completement” to “1. hydrous”. (I think especially because the coding examples do expect you to read more in that manner.)

**Reasons To Accept:**

This is a thoughtful piece of work that begins with a clear presentation of the topic that the paper will explore. It makes analogy to concepts from psychology (while walking a reasonable line not to make assumptions that the model behavior is identical, just analogous in some ways) and uses clear examples to set the reader up to understand the main concepts before following up with clear and careful definitions.

The experimental setup is clear and generally seems appropriate. I found that the writing did a good job of anticipating and addressing questions I was likely to have as a reader, such as controlling for word length and exploring casing-related tasks (I was thinking of some issues that arose in https://aclanthology.org/D18-1339.pdf, though that examined other copying-related topics, with pre-Transformer architectures).

The work does a good job of balancing the broader picture through results across full datasets and giving the reader specific qualitative examples.

**Reasons To Reject:**

There are some areas (see questions below) that could use additional clarification. There are also some areas that are quite dense; it may be appropriate to provide the readers slightly more detail on approaches like patching as they are introduced.

The claim that “Concept Induction is Language-Agnostic” may be somewhat too strong; the languages evaluated in this are all fairly high-resource (though this is somewhat acknowledged in the cited work). It would be helpful to do additional evaluation with very low-resource languages. That said, this is beyond the scope of this paper to fully explore, and I think publishing this paper is likely to inspire future work that takes a similarly careful look at this question.

The section on Function Vector Heads feels like it arrives quite suddenly and it would be nice to have an expanded analysis of the relations between the different types of heads explored, given additional space.

---

> ### Author Response · Authors · 2025-05-28
>
> Thank you for your detailed and helpful review! We’re working on updating our paper to address all of your questions and comments. We hope that these fixes will provide some clarity for sections of the paper that feel dense right now.
>
> We also thank you for the constructive feedback on our last two sections.
> - You bring up a good point about how we only evaluate on high-resource languages. It’s hard to make the strong claim we do about representations being “agnostic” to language when we only look at a small subset of languages. This almost surely depends on a model’s exposure to any given language. We will adjust our claims in Section 5 in the final version, if accepted.
> - We agree that the FV section might feel a little sudden. Our intuition is that FV heads and concept induction heads are similar mechanisms that play distinct roles: FV heads can be thought of as a “sister” to concept induction heads. However, we don’t state this very clearly in the text. We will update the manuscript to make this idea clearer.

---

> > ### Comment · Reviewer_3CWa · 2025-06-03
> >
> > As my main concerns with this paper are presentational (just being clear about the limitations of evaluating on high-resource languages, being cautious with phrases like "language-agnostic, and providing a bit more guidance to the reader), I expect I would find these small adjustments satisfactory. I think that simply giving a little more context to the FV section will make it feel less abrupt to readers.
> >
> > To confirm that I don't have a misunderstanding, the one question from the "Questions to Authors" section that it would be helpful to get a quick answer to here is: Lines 258-260: “FV heads cause the model to output the same concept in a different language, whereas concept induction heads cause the model to output a different concept in the same language” – should “patching” appear somewhere in this sentence?

---

> > ### Author Response · Authors · 2025-06-04
> >
> > Thank you for the reply! We definitely do not want to over-claim, and have done our best to adjust the manuscript to address your concerns, which we agree with. Particularly, in our "language-agnostic" section, we've added some more discussion:
> > > ... This is in line with previous work suggesting that concept information in LLMs may be language-agnostic~\citep{wendler2024llamas, dumas2024separating, brinkmann2025large}, though Schut et al. (2025) argues concepts for some tasks may be biased towards English. **For model representations to be truly language-agnostic, they must be trained on those languages---thus, even if high-resource languages are represented in a unified semantic space, this effect may not hold for low-resource languages. Unfortunately, as we do not evaluate on low-resource languages in this work, we cannot make claims as to how models represent low-resource languages.**
> >
> > And yes, "patching" should appear in that sentence (which we have now updated). Now, it reads:
> > > Figure 7 shows that for the exact same prompts, patching FV heads causes the model to output the same concept in a different language, whereas patching concept induction heads causes the model to output a different concept in the same language.
> >
> > Your other notes were also helpful. e.g., 73-74 we did mean sampling as described in the footnote, which we have also changed:
> > > which we corrupt by replacing the first half with different random tokens: $p_{corrupt}=y_1y_2...y_{n+m}|x'_1x'_2...x'_{n}c'_1$.
> >
> > Thanks so much, and please let us know if you have any other concerns.

---

### Decision · Program_Chairs · 2025-07-08

**Decision:**

Accept

**Comment:**

This paper received mixed reviews: two very positive ones (including a reviewer fighting for acceptance) and two somewhat negative ones.

Synthesizing the reviews and my reading, I believe the paper has the following relevant strengths and weaknesses.

Reasons To Accept:
* The paper was described as well-written at various levels by the reviewers [3CWa,ZnwZ]
* Extensive ablation studies [ZnwZ]
* Consistent results across different LMs [ZnwZ]
* The paper provides foundational insights into in-context learning [sPEp,NwaB]

Reasons To Reject:
* The paper was perceived as dense and not sufficiently clear in some technical regards [3CWa]. I urge the authors to do what they can to improve this aspect.

There are some other listed “reasons to reject” that I believe do not pose major concerns:
* The patching effect might be not very large [sPEp], and it’s not evident how the results help applications [NwaB]. I consider this fine for foundational research about language models.
* Evaluation relies on synthetic or narrow tasks [NwaB]. I concur with the authors that such tasks are commonly used to study ICL, and their constrained nature can help get a clean picture.

Taken together -- and despite the mixed scores provided by the reviewers -- I believe, taking together the overall set of strengths and weaknesses, the ensuing discussion, and my reading of the paper, that the paper is of sufficiently high quality to merit publication in COLM.